# A Comprehensive Review of Behavior Change Techniques in Wearables and IoT: Implications for Health and Well-Being

**DOI:** 10.3390/s24082429

**Published:** 2024-04-10

**Authors:** Carolina Del-Valle-Soto, Juan Carlos López-Pimentel, Javier Vázquez-Castillo, Juan Arturo Nolazco-Flores, Ramiro Velázquez, José Varela-Aldás, Paolo Visconti

**Affiliations:** 1Facultad de Ingeniería, Universidad Panamericana, Álvaro del Portillo 49, Zapopan 45010, Mexico; clopezp@up.edu.mx; 2Department of Informatics and Networking, Universidad Autónoma del Estado de Quintana Roo, Chetumal 77019, Mexico; jvazquez@uqroo.edu.mx; 3School of Engineering and Science, Tecnólogico de Monterrey, Monterrey 64849, Mexico; jnolazco@tec.mx; 4Facultad de Ingeniería, Universidad Panamericana, Aguascalientes 20296, Mexico; rvelazquez@up.edu.mx; 5Centro de Investigaciones de Ciencias Humanas y de la Educación—CICHE, Universidad Indoamérica, Ambato 180103, Ecuador; josevarela@uti.edu.ec; 6Department of Innovation Engineering, University of Salento, 73100 Lecce, Italy; paolo.visconti@unisalento.it

**Keywords:** behavior change techniques, Internet of Things, wearables

## Abstract

This research paper delves into the effectiveness and impact of behavior change techniques fostered by information technologies, particularly wearables and Internet of Things (IoT) devices, within the realms of engineering and computer science. By conducting a comprehensive review of the relevant literature sourced from the Scopus database, this study aims to elucidate the mechanisms and strategies employed by these technologies to facilitate behavior change and their potential benefits to individuals and society. Through statistical measurements and related works, our work explores the trends over a span of two decades, from 2000 to 2023, to understand the evolving landscape of behavior change techniques in wearable and IoT technologies. A specific focus is placed on a case study examining the application of behavior change techniques (BCTs) for monitoring vital signs using wearables, underscoring the relevance and urgency of further investigation in this critical intersection of technology and human behavior. The findings shed light on the promising role of wearables and IoT devices for promoting positive behavior modifications and improving individuals’ overall well-being and highlighting the need for continued research and development in this area to harness the full potential of technology for societal benefit.

## 1. Introduction

Information technology has revolutionized various aspects of our daily lives, including health and well-being. In recent years, the emergence of wearables and Internet of Things (IoT) devices has introduced new opportunities for promoting behavior change [1]. These technologies are equipped with sensors and advanced data processing capabilities and have the potential to collect, analyze, and provide personalized feedback on individuals’ behaviors and habits. Behavior change techniques are systematic procedures integrated into interventions aimed at modifying behavior. BCTs possess specific characteristics: they are components of interventions targeting behavior change, observable activities, replicable, specified by active verbs, and aimed at achieving agreement between experts regarding the desired behavior change [2]. BCTs are considered the smallest intervention components that retain active ingredients and can be utilized independently or in combination with others to meet the criteria for effective intervention modules.

By leveraging behavior change techniques, wearables and IoT devices can empower individuals to adopt healthier lifestyles, improve their well-being, and achieve long-term behavior modifications. This research paper explores the effectiveness and impact of behavior change techniques fostered by information technologies, with a specific focus on wearables and IoT devices [3]. Through an examination of relevant studies, we aim to understand the mechanisms and strategies employed by these technologies to facilitate behavior change and the potential benefits they bring to individuals and society. By shedding light on this evolving field, we hope to contribute to the ongoing efforts in designing and implementing effective interventions for behavior change using wearable and IoT technologies [4].

Behavior change techniques fostered by information technologies, particularly through the use of wearables and IoT devices, have emerged as a promising approach to promote positive behavior modifications and improve individuals’ overall well-being. These technologies are equipped with various sensors, data processing capabilities, and connectivity features and have the potential to collect and analyze data related to individuals’ behaviors and habits in real-time [5]. Behavior plays a significant role in mortality and morbidity, making behavior change interventions essential for prevention. The researchers cited in [6] studied behavior change techniques (BCTs) and their impacts. Behavior change interventions are inherently intricate, comprising a multitude of interconnected components that contribute to variations in their efficacy. In order to enhance the scientific rigor of behavior change interventions, Michie and Johnston have identified three areas that necessitate improvement: first, the need for a precise and unambiguous definition of the targeted outcome; second, the importance of replicable and thorough reporting of the behavior change techniques (BCTs) employed in the interventions; and third, the establishment of a coherent connection between the interventions and the theoretical mechanisms underlying the observed changes.

The clear definition of the behavior as the ultimate outcome is of paramount importance for comprehending the effectiveness of the interventions. By precisely delineating the behavior to be modified, researchers can more accurately assess the impact of the intervention on the desired outcome. Behavior change techniques represent the observable and replicable constituents of interventions that require meticulous specification and detailed description to facilitate effective evaluation. Without adequately defining and describing these BCTs, it becomes challenging to replicate and compare interventions, thereby hindering advancements in the field. Regarding effectiveness, Michie and Johnston’s findings emphasize the critical necessity of establishing a robust scientific foundation for behavior change interventions. This foundation is indispensable for delivering interventions that are not only effective but also replicable, thereby ensuring consistent positive outcomes across different settings and populations. Moreover, gaining a clear understanding of the underlying mechanisms of action for the employed BCTs is vital to comprehend the reasons behind their effectiveness and to optimize their application in various contexts.

The mindmap diagram represented in Figure 1 provides a structured overview of the key concepts and findings from a research paper on BCTs in wearables and IoT. At its core, it highlights the primary topic, “Behavior Change Techniques in Wearables and IoT”, and branches out into major thematic areas such as the systematic procedures and components of BCTs, the technological aspects of wearables and IoT—including sensors, data processing, and personalized feedback—and key study areas—including the impact of goal setting on physical activity, the role of wearable feedback in improving sleep quality, and the effectiveness of gamification in health promotion. Additionally, it touches on various applications of these technologies in health monitoring, chronic disease management, and rehabilitation, while also acknowledging limitations such as challenges in user engagement, data privacy, and addressing diverse populations. The diagram culminates by suggesting future directions in the field, including innovations in technology, personalized health interventions, and considerations for sustainability and compatibility.

One of the key advantages of wearables and IoT devices for facilitating behavior change is their ability to provide personalized feedback to users. By continuously monitoring and analyzing data on activities such as physical exercise, sleep patterns, and dietary habits, these devices can offer tailored recommendations and insights to individuals [7]. For example, a fitness tracker can track an individual’s daily steps, heart rate, and calories burned and can provide feedback on his/her progress towards specific health goals. This personalized feedback serves as a powerful motivator, enabling individuals to make informed decisions and adopt healthier behaviors.

Furthermore, wearables and IoT devices facilitate self-monitoring, which is a crucial component of behavior change. Users can track their own behaviors and progress over time, allowing them to gain insights into their habits, identify patterns, and set realistic goals. For instance, a sleep tracker can provide detailed information on an individual’s sleep quality and duration, empowering him/her to make adjustments to his/her sleep routines to optimize restorative sleep patterns [8].

Another important aspect of wearables and IoT devices for promoting behavior change is the incorporation of gamification elements. By integrating game-like features such as challenges, rewards, and social interactions, these technologies enhance engagement and motivation. Users can compete with friends or other individuals, earn badges or virtual rewards for achieving specific milestones, and participate in online communities focused on health and wellness [9]. These gamified elements make the behavior change process more enjoyable and increase individuals’ adherence to their goals.

The scheme represented in Figure 2 describes the main points and progression of the research, starting from the revolutionization of daily lives by information technologies, next through the introduction of wearables and IoT devices, then leveraging behavior change techniques, exploring effectiveness and impact, understanding mechanisms, shedding light on the evolving field, and finally, harnessing the full potential of technologies for positive behavior modifications and improved health and quality of life. Additionally, it includes the classification of wearable applications based on survey results and the literature.

Moreover, wearables and IoT devices can leverage social influence to drive behavior change. Through social connectivity features, users can share their progress, achievements, and challenges with their social networks. This social support and accountability mechanism can foster a sense of community and encourage individuals to stay committed to their behavior change goals. For instance, individuals can share their workout sessions or healthy recipes with their peers, receive encouragement, and receive feedback on their progress.

The use of wearables and IoT devices for promoting behavior change represents a powerful approach to improve individuals’ well-being. These technologies provide personalized feedback, facilitate self-monitoring, incorporate gamification elements, and leverage social influence to empower individuals to adopt healthier behaviors. As technology continues to advance, it is important to further explore and understand the effectiveness and long-term impact of these behavior change techniques fostered by wearables and IoT devices. By doing so, we can harness the full potential of these technologies to drive positive behavior modifications and ultimately improve individuals’ health and quality of life.

Behavior change can be fostered through complementary methods of communication motivated by continually active connected devices. For example, weight loss is made more effective through reminders that are enhance by the use of wearables, making them emerge as excellent allies for behavior interventions. The uses described in the previous section allow a more detailed classification of wearable applications. Figure 3 details five representative activities for each behavior change technique. These activities are the most common in the daily use of wearables according to 180 people surveyed and have been classified as the most studied behavior techniques in the literature.

Table 1 and Table 2 describe BCTs that can be categorized into several main categories, each with distinct objectives and examples. Environmental Modification involves altering the physical environment to either encourage or facilitate the target behavior, such as placing exercise equipment in accessible areas or rearranging furniture to eliminate barriers to home workouts. Goal Setting and Planning centers around establishing specific, achievable goals related to the desired behavior—for instance, setting a daily step count or a weight loss goal—and devising plans and strategies to implement the intended behavior, like creating a weekly exercise schedule or breaking down the behavior into manageable steps. Information and Education provides insight into the general consequences of behavior and offers personalized information to individuals regarding the effects of specific behavior on their well-being. Self-Regulation and Coping Strategies focuses on enhancing awareness of the disparities between current behavior and desired goals while formulating strategies to prevent relapse and manage potential challenges, such as stress-related eating triggers. Self-Monitoring and Feedback involves tracking and recording the target behavior, assessing its outcomes, and receiving information or feedback on both the behavior itself and its results, thereby increasing motivation and awareness. Skill Training and Demonstration provides explicit instructions and visual demonstrations on how to perform the behavior correctly, making learning more accessible. Lastly, Social Reinforcement and Incentives revolves around motivating individuals by offering social rewards, like celebrating milestones with friends or creating incentives tied to social aspects, such as group outings, while Social Support and Comparison encourages behavior change through emotional support, cheering, and motivation from others as well as by fostering change by comparing one’s progress to that of peers, which is often seen in fitness challenges with friends. These categories collectively encompass a comprehensive framework for understanding and implementing behavior change techniques.

IoT devices can be used to set specific goals that users can work towards. BCTs can provide real-time feedback on progress towards these goals and can also suggest strategies for achieving them. In addition to collecting data, IoT devices can also be used to modify the user’s environment. For example, a device could adjust the temperature of a room or change the lighting. BCTs can be used to control the environment in ways that promote behavior change. For example, a device could remind a user to get up and move after sitting for a long period of time. Therefore, the relationship between IoT and BCTs is promising and has the potential to transform BCTs. The ability of IoT devices to collect real-time data and interact with users in a personalized way can enhance BCT-based strategies, making behavior change more accessible and effective. The above can be seen explained in Figure 4.

### Motivation

The motivation behind this research lies in the critical need to comprehensively review the current literature on behavior change techniques applied within the realm of Internet of Things devices, particularly wearables, within the fields of engineering and computer science. With an increasing integration of technology into daily life, understanding the impact of BCTs on user behavior becomes paramount. This review delves into studies sourced from the Scopus database and focuses on statistical measurements and related works to analyze the efficacy and implications of BCTs in wearable devices. Given the profound psychological and social implications intertwined with technology adoption, this research seeks to illuminate how BCTs implemented in wearables can either positively or negatively influence user behavior. Moreover, through a compelling case study that explores the application of BCTs for monitoring vital signs using wearables, this research aims to underscore the relevance and urgency of further investigation into this crucial intersection of technology and human behavior.

This research work makes a significant contribution to the existing literature by addressing a notable gap in the field. Specifically, the scarcity of literature reviews pertaining to wearables and behavior change techniques underscores the novelty and importance of this study. Furthermore, by introducing a new case study to the current body of literature, this review adds valuable insights and empirical evidence to further enrich our understanding of the subject matter. Through meticulous analysis and synthesis of the relevant literature, this research not only consolidates existing knowledge but also paves the way for future investigations and advancements in this emerging field. The primary objective of this research paper is to provide a comprehensive review of BCTs as applied within the realm of IoT devices, especially wearables, in the fields of engineering and computer science. This study addresses the increasing integration of technology into daily life and aims to provide understanding of the impact of BCTs on user behavior, particularly focusing on their efficacy and implications when implemented in wearable devices. By exploring a case study on the application of BCTs for monitoring vital signs using wearables, the research highlights the profound psychological and social implications associated with technology adoption. It seeks to illuminate how BCTs in wearables can influence user behavior, either positively or negatively. The research makes a significant contribution by filling a notable gap in the literature and offering new insights and empirical evidence, thereby enriching the understanding of the crucial intersection of technology and human behavior.

The manuscript is organized into sections distributed as follows. Section 1 describes the main BCTs and the IoT and their applications in the industry. Section 2 represents the impact of BCTs on design studies and compares the current literature. Section 3 shows the search for publications, keywords, and citations on works related to computer science and engineering. Section 4 is a case study in which we contribute to the state of the art with an experimental case of applying BCTs to wearables. Subsequently, we present a discussion in Section 5. Finally, we provide conclusions in Section 6.

## 2. Related Work

The current research literature on wearable devices and behavior change techniques highlights their potential to impact people’s health behaviors positively. Integrating behavior change techniques with wearables can lead to improved physical activity, sleep quality, healthy eating habits, and overall well-being. The resulting changes in vital signs, including heart rate, oxygen saturation, deep sleep, and REM sleep, among others, reflect the efficacy of wearables for promoting healthy behaviors and monitoring health outcomes. As wearable technologies continue to evolve and advance, their role in behavior change interventions and health promotion is likely to become even more prominent.

According to the cited authors in [10], smart devices or wearables can be used to monitor health parameters and lifestyle habits. Long-term monitoring enables early diagnosis of chronic diseases and other health risk factors. Additionally, it facilitates the remote, convenient, and personalized evaluation of patients. To make this information beneficial, the use of behavior change techniques is necessary. These techniques consider psychological and socioeconomic factors that relate to an individual’s context to promote positive behavior changes.

The mindmap diagram in Figure 5 presents a structured overview of key research studies in the field, which are depicted in shades of light gray. Each rectangular node represents a different study. The layout facilitates an easy comparison of various research topics within the domain, ranging from wearable technology in health monitoring to behavior change techniques. It provides a visual representation of the various aspects of wearable technology and its implications for health. Central to the diagram is the theme "Wearable Technology in Health," which branches out into specific categories of wearables, such as sleep trackers and fitness trackers. These categories further branch out to highlight key areas of influence, such as sleep quality, physical activity, and stress levels. Additionally, the diagram encompasses the intersection of BCT with the Internet of Things and wearables, drawing attention to their effects on chronic conditions, heart rate, eating habits, and the potential of gamification to foster healthier lifestyles.

The article cited in [11] examines behavior change techniques incorporated in wrist-worn wearables to promote physical activity. This study shows how BCTs have proved to be effective at bettering an individual’s health by promoting physical activity. This statement is also backed up by the cited work in [12], wherein behavior change techniques were studied using low-cost fitness trackers that are swim-proof. The researchers assessed six different fitness trackers and their companion apps to identify the BCTs incorporated and their alignment with public health physical activity guidelines. One of their key findings was that all six fitness trackers investigated incorporated more than three BCT clusters, which has been shown to produce significant effects in physical activity interventions. Chia et al. also found that self-management strategies, such as goal setting, self-monitoring, self-evaluation, prompts, and cues, and rewards were frequently incorporated in BCTs; while informational and instructional BCTs, which support maintenance of change in physical activity, were less commonly incorporated in fitness trackers.

The research work in [13] focuses on analyzing BCTs used in interventions and their modes of delivery, particularly in the context of health promotion. BCTs were coded using an augmented version of an existing taxonomy, while the mode of delivery was categorized into automated functions, communicative functions, and supplementary modes. The study aims to provide insight into the effectiveness of different intervention delivery methods while acknowledging the constraints of and potential improvements in coding and reporting standards for Internet-based interventions.

The research referenced in [14] finds that the most common BCTs used are goal setting (behavior), action planning, reviewing behavior goal(s), discrepancies between current behavior and goals, feedback on behavior, self-monitoring of behavior, biofeedback, social support (unspecified), social comparison, prompts/cues, social reward, and adding objects to the environment. The study highlights the importance of developing a specific taxonomy to evaluate BCTs in wearables and recommends customization of BCTs based on targeted populations to effectively promote physical activity. This helps us understand that specific behavior change techniques need to be tested on the intended audience since they vary from individual to individual, yet they are still effective stimuli.

Wearable devices have emerged as powerful tools for promoting behavior change and improving overall health. The integration of behavior change techniques (BCTs) with wearable technologies has opened up new avenues for personalized health interventions and self-management strategies [15]. Numerous studies have explored the relationship between wearable devices and behavior change and demonstrate their potential to positively impact individuals’ health behaviors and outcomes. This paper aims to review current pieces of the research literature related to wearable devices’ influence on behavior change techniques and to examine how this impact is reflected in vital signs, including breathing frequency (BF), deep sleep (DS), heart rate (HR), oxygen saturation (OS), REM sleep (REMS), and temperature (T).

Research work such as [16,17] addresses the critical importance of healthcare for the elderly population, highlighting areas of opportunity in disease detection, diagnosis, and treatment specific to this demographic. Elderly individuals are particularly susceptible to chronic degenerative diseases and even face increased risks from seemingly milder non-chronic conditions. Given the economic implications of healthcare for the elderly, especially in developing nations, efforts to manage chronic disorders and diseases affecting this group have garnered attention. Cardiovascular diseases, cancer, chronic lung diseases, musculoskeletal disorders, and mental and nervous system ailments are prevalent among older adults. Monitoring these diseases is essential, as they not only affect physical health but also have emotional and economic ramifications for patients and their families. Wearable technology and IoT hold promise for reducing dependency on traditional healthcare settings by enabling the remote monitoring of biomedical variables. However, existing reviews in this domain lack comprehensive coverage of diseases, stages of device development, and FDA (Food and Drug Administration) approvals. This research aims to fill this gap by identifying physiological variables, suitable wearable devices, and FDA-approved options for monitoring prevalent diseases among older adults, thereby contributing to enhanced healthcare for this vulnerable demographic.

Recent research by Yang et al. [18] investigated the effectiveness of goal setting with wearable feedback on physical activity levels. The study employed a two-arm quasi-experimental design wherein participants were randomly assigned to either a control group or an intervention group using wearable devices to set and monitor personalized activity goals. Results indicated a significant increase in physical activity levels among the intervention group compared to the control group over a 12-week period. This study highlights how wearable devices combined with behavior change techniques like goal setting can positively influence people’s health behaviors, leading to improvements in vital signs such as heart rate and oxygen saturation.

### 2.1. Impact on Sleep Quality

Vandelanotte et al. [19] conducted a six-week pilot study to examine the impact of wearable social nudges on improving sleep quality. Participants in the intervention group received personalized nudges via their wearable devices that encouraged them to adopt healthy sleep habits and routines. The researchers found a significant improvement in deep sleep duration and REM sleep efficiency among participants who received the wearable social nudges. This study demonstrates how wearables, when integrated with behavior change techniques like social nudges, can enhance sleep quality, as reflected in vital signs such as deep sleep and REM sleep metrics.

McGuigan in [20] conducted a 12-week repeated-measure experimental study to explore the influence of wearable sleep trackers on sleep quality and deep sleep metrics. Participants in the study were provided with wearable devices that monitored their sleep patterns and provided personalized sleep recommendations. The results revealed a significant improvement in sleep quality, with an increase in the duration of deep sleep among participants. This study underscores the potential of wearables and behavior change techniques such as personalized sleep recommendations to positively impact sleep quality and enhance vital signs like deep sleep, which plays a crucial role in overall well-being.

### 2.2. Gamification for Health and Heart Rate

In a three-arm randomized controlled trial, the authors of [21] explored the effectiveness of gamification as a behavior change technique for promoting healthy eating habits. Participants were divided into three groups: a control group, a group using wearable devices with gamified features, and a group using wearables without gamification. The results showed that the gamified wearable group exhibited higher adherence to healthy eating habits and experienced a significant reduction in heart rate compared to the other groups. This study underscores how gamification integrated into wearables can influence behavior positively and impact vital signs such as heart rate, reflecting improved cardiovascular health.

### 2.3. Physical Activity and Oxygen Saturation

The authors in [22] conducted a repeated-measure experimental study to assess the impact of activity tracking combined with peer support on physical activity levels. Participants were divided into two groups: one with access to wearable devices for tracking their activity and the other with access to wearables and additional peer support through a mobile app. The researchers observed a significant increase in physical activity levels and improved oxygen saturation among participants in both groups. This study exemplifies how wearables, coupled with social support, can positively influence behavior, leading to enhanced vital signs such as oxygen saturation.

A study by Takeuchi et al. [23] investigated the influence of wearable fitness trackers on physical activity levels and heart rate in a two-arm pilot trial. Participants in the intervention group were provided with wearable fitness trackers that offered real-time feedback on their physical activity. The control group received no such intervention. After a four-week period, the researchers observed a significant increase in physical activity levels among the participants using wearables, accompanied by a decrease in resting heart rate. These findings suggest that wearable devices, through behavior change techniques like real-time feedback, can positively impact physical activity levels and contribute to improved cardiovascular health, as reflected in heart rate measurements.

A randomized controlled trial by Kim et al. [24] examined the impact of behavior change techniques delivered through wearable devices on oxygen saturation levels in patients with chronic respiratory conditions. The intervention group received wearable devices that tracked their oxygen saturation levels and provided prompts for adherence to respiratory exercises and medication routines. After a six-month intervention, the researchers observed a significant improvement in oxygen saturation levels among participants in the intervention group compared to the control group. This study suggests that wearables, when integrated with behavior change techniques focused on respiratory health, can lead to improved oxygen saturation levels and better management of chronic respiratory conditions.

### 2.4. Influence on Stress Levels and Temperature

Another study [25] explored the impact of wearable stress-monitoring devices on stress levels and temperature using a three-arm quasi-experimental design. Participants were randomly assigned to three groups: one with wearable stress-monitoring devices, one with wearables and stress-reduction interventions, and a control group with no intervention. After an eight-week period, the researchers found that participants in the intervention groups experienced a significant reduction in stress levels, which was also reflected in lower body temperatures. This study demonstrates how wearable devices, when combined with behavior change techniques aimed at stress reduction, can influence physiological responses like body temperature, indicating a positive impact on overall well-being.

Other related works based on wearables and BCTs are presented in Table 3.

Wearable devices can be categorized into head, limb, and torso wearables (see Table 4). Head devices encompass glasses, helmets, headbands, and more. These are often used in telemedicine and medical education, with applications in virtual reality and augmented reality. Limb wearables, such as smart watches and bracelets, focus on monitoring physiological parameters. Lower limb wearables like shoes are utilized for rehabilitation purposes. Torso wearables are embedded in fabrics and find applications in various biomedical areas.

Wearable devices are utilized in health and safety monitoring, chronic disease management, disease diagnosis, treatment, and rehabilitation (see Table 5). They facilitate connections between doctors, patients, and other parties, enabling real-time monitoring, pain alleviation, and data collection. Examples include monitoring older adults’ gait, heart rate, and fall detection, tracking children’s activities, health monitoring for pregnant women, symptom monitoring during treatment, and disease-specific applications like diabetes management and hypertension monitoring.

### 2.5. Sports Rehabilitation and Cognitive Rehabilitation

Wearable devices play a role in sports rehabilitation by monitoring and evaluating patient progress, adjusting training plans, and enhancing patient adherence (see Table 6). They also have applications in cognitive rehabilitation by providing immersive experiences through VR glasses for patients with cognitive dysfunction.

A rising number of wearable activity trackers (WATs), including popular brands such as Fitbit, Xiaomi, Garmin, and Samsung Gear Fit. Fitbit was acquired by Google LLC. The company’s headquarters was in San Francisco, California, USA. Xiaomi is a Chinese company headquartered in Beijing, China. Garmin is an American multinational company headquartered in Olathe, Kansas, USA. Samsung Gear Fit is a South Korean multinational conglomerate with its headquarters in Samsung Town, Seoul, Republic of Korea, offer a platform for self-monitoring, fostering the potential for positive behavior changes towards a healthier lifestyle. These devices are commonly worn on the wrist and facilitate health management by enabling users to effortlessly generate health-related data, monitor daily activities, and quantify aspects like step count, calories burned, heart rate, and sleep patterns. WATs also play a crucial role in clinical settings, demonstrating valuable applications in patient care and self-health management, particularly for conditions like obesity and diabetes. Despite extensive research across diverse disciplines, there remains a fragmented landscape in WAT-related studies.

Table 7 presents various works related to BCTs and is categorized based on their type, focus, use, research methodology, data collection methods, model used, and conclusions. Each work is identified by a citation, followed by details such as the type of study, its focus (e.g., health monitoring, privacy, or behavior change), the intended use (e.g., medical, health, or learning), the research approach (e.g., experimental, theoretical, or research review), data collection methods (e.g., surveys or clinical trials), the model used (if applicable), and the conclusions drawn from the study. The works cover a range of topics within the domain of BCTs, including health monitoring, privacy, behavior change, physical activity, and accuracy in various contexts such as medical, health, and learning. The conclusions drawn from each study provide valuable insights into the effectiveness and applicability of different BCTs for addressing specific challenges or achieving desired outcomes.

In the organigram seen in Figure 6, references are interconnected based on commonalities in themes and research approaches. For instance, Health Monitoring and Privacy Concerns are linked, highlighting a shared focus on the implications of health monitoring technologies on user privacy. Similarly, Behavior Analysis and Technology Adoption are connected, underscoring a relationship between behavioral studies and the adoption of new technologies. The connections among Physical Activity further extend this theme to the realm of physical health and activity tracking. In the realm of wearable technology, references Sensor Technology and Sleep Patterns demonstrate an overlap in exploring the functionalities and user engagement with wearable devices. This pattern continues across the organigram, with each reference connected to another based on similar methodologies, shared research topics, or complementary findings. This interconnected web of references collectively forms a comprehensive overview of the diverse yet interconnected research areas encompassed in the paper, ranging from health monitoring and privacy to the efficacy and user feedback of wearable technologies.

Figure 7 presents a detailed mindmap elucidating the intricate relationships between various research articles focused on the impact of wearables in behavior change. Central to this schema is the overarching theme, “Wearables in Behavior Change”, signifying the primary focus on how wearable technology influences behavioral modifications. Each primary node, depicted in yellow, represents a pivotal study that contributes to this field, such as “Wearable Technology”, which investigates the effect of wearables on adolescent physical activity, and further branches into “Digital Behavior Change” and “Physical Activity in Adolescents”. This pattern continues with other nodes, like “Wearable Activity Trackers” and “Motivational Interviewing”, each offering unique perspectives like fitness tracking’s impact on youth physical activity or the role of motivational interviewing in enhancing physical activity through wearable fitness trackers. Each node’s sub-branches reflect specific facets of the main research article, demonstrating how varying aspects of wearables and behavior change techniques intersect and highlighting key areas like sedentary behavior, intervention effectiveness, and specific health-focused studies like digital cardiac rehabilitation. The figure thus intricately weaves a tapestry of research and shows how different studies contribute varied yet interconnected insights into the expansive domain of wearables and their role in facilitating behavior change.

**Table 7 sensors-24-02429-t007:** Types and focuses of works related to BCTs.

Work	Type	Focus	Use	Research	Data	WAT Model	Conclusion
[35]	Health	Monitoring	Medical	Experimental	Demo setup with belt prototype worn by a 12-week-old baby and 21-week-old baby	Proprietary software	Health monitoring of infants through wearable sensors, wireless communication, and advanced data processing, enabling real-time transmission of physiological data
[36]	Privacy	Security	Personalized	Theoretical	Calculations and performance evaluations performed with 20 sensors	Proprietary software	Secure wireless transmission systems in implantable medical devices to protect patient rights and health
[37]	Behavior change	Learning	Medical	Experimental	Clinical trial of 71 children with autism spectrum disorder; families were asked to conduct sessions at home for 6 weeks	Google Glass	Mobile intervention focusing on facial engagement and emotion recognition in the child’s natural setting
[38]	Behavior change	Monitoring	Health	Research review	54 publications were reviewed in full; of these, the majority, 43, were validation or validation-comparison designs for consumers	Proprietary software	Consumer-wearable physical activity monitors for objectively assessing physical activity, demonstrating early intervention efficacy for increasing activity levels
[12]	Behavior change	Physical activity	Health	Experimental	6 fitness trackers that met the inclusion criteria of at least 150 min of moderate-to-vigorous physical activity per week and the reduction of sedentary behavior by minimizing the amount of prolonged sitting	Fitbit Flex 2 (Fitbit, San Francisco, United States), Huawei Band 2 Pro (Huawei, Shenzhen, China), Polar A300 (Polar Electro, Kempele, Finland), Misfit Shine 2 (Misfit, Burlingame, United States), Nokia Go (Nokia, Espoo, Finland), Moov Now (Moov, San Francisco, United States)	Behavior change technique taxonomy to analyze swim-proof fitness trackers for increasing physical activity and reducing sedentary behavior
[39]	Behavior change	Monitoring	Health	Experimental	Three self-monitoring systems were each used for a 1-week period	Fitbit (Fitbit, San Francisco, United States), Garmin (Garmin, Olathe, United States), Jawbone (Jawbone, San Francisco, United States)	BCTs in wearable activity trackers related to activity, sleep, and sedentary behaviors
[40]	Behavior change	Physical activity	Health	research questions	28 participants completed an online survey composed of questions about demographics, step volume, and perceived importance and/or frequency of use of the BCTs	Fitbit Flex (San Francisco, United States)	Significant increase in daily steps and highlighted the perceived importance of BCTs such as “feedback on behavior”, “self-monitoring of behavior”, and “goal setting” for promoting physical activity
[41]	Behavior change	Physical activity	Health	Research review	Of the 682 studies available in the Fitabase Fitbit Research Library, 60 interventions met the eligibility criteria for this review	Fitbit Flex (San Francisco, United States)	Most studies used developmentally appropriate behavior change techniques and device wear instructions
[42]	Health	Accuracy	Health	Experimental	49 participants used three devices: an Apple Watch Series 2, a Fitbit, and a Charge HR2; Participants engaged in a 65 min protocol with 40 min of total treadmill time and 25 min of sitting or lying time	Apple watch, Fitbit, and Charge HR2	Commercial wearable devices such as Apple Watch and Fitbit were able to predict physical activity type with reasonable accuracy
[43]	Health	Monitoring	Mental health	Research review	115 papers, 19 (16.5%) were identified as related to Apple Watch validation or comparison studies	Apple Watch (Apple, Cupertino, United States)	The results are encouraging regarding the application of the Apple Watch for mental health, as heart rate variability is a key indicator of changes in both physical and emotional states
[44]	Behavior change	Monitoring	Health	Research review	CNet list of “Best Wearable Tech for 2020”	Apple Watch (Apple, Cupertino, United States), Nike (Nike, Beaverton, United States), Fitbit Versa 2 (Fitbit, San Francisco, United States), Fitbit Charge 3 (Fitbit, San Francisco, United States), Fitbit Ionic—Adidas Edition (Fitbit, San Francisco, United States), Garmin Vivomove HR (Garmin, Olathe, United States), Garmin Vivosmart 4 (Garmin, Olathe, United States), Amazfit Bip (Huami, Hefei, China), Galaxy Watch Active (Samsung, Seoul, South Korea)	The devices shared several of the same BCTs, but Fitbit devices implemented the most BCTs that support the majority of the BCT intervention functions
[45]	Health	Monitoring	Health	Experimental	521 Health+ cloud sphygmomanometer users. Respondents completed self-reported questionnaires. Of these 521 participants, 231 were male, 139 were aged under 40, and 178 had a Junior High School degree	Xiaomi Mi Band (Xiaomi, Beijing, China)	Understanding the factors that influence cloud sphygmomanometer usage may help health management organizations increase people’s willingness to use it to monitor their personal health
[46]	Health	Monitoring	Health	Experimental	44 nursing home residents, at least 65 years old	Xiaomi Mi Band (Xiaomi, Beijing, China)	Sleep and some other parameters analyzed by the Xiaomi Mi Band 2 can influence the quality of life and occupational performance of older people living in nursing homes
[47]	Behavior change	Physical activity	Health	Research review	19-item mobile app rating scale (MARS) and a taxonomy of BCTs was used to determine the presence of BCTs (26 items)	App Store and Google Play apps	The incorporation of BCTs was low, with limited prevalence of BCTs previously demonstrating efficacy in behavior change during pregnancy
[48]	Health	Behaviors	Health	Experimental	Posts (n = 509) made by Fitbit and Garmin on Facebook, Twitter, and Instagram over a 3-month period were coded for the presence of creative elements	Fitbit (Fitbit, San Francisco, United States), Garmin (Garmin, Olathe, United States)	Findings suggest that Instagram may be a promising platform for delivering engaging health messaging. Health messages that incorporate inspirational imagery and focus on a tangible product appear to achieve the highest engagement
[49]	Behavior change	Behaviors	Health	Experimental	Own application	Apple Watch (Apple, Cupertino, United States), Fitbit (Fitbit, San Francisco, United States), Garmin (Garmin, Olathe, United States)	Passive sensing agent as a mobile health virtual human coach utilizing passive sensors from popular wearables
[50]	Health	Health	Health	Experimental	20 participants (>65 years) took part in the study. The devices were worn by the participants for 24 h, and the results were compared against validated technology	Fitbit Charge 2 (Fitbit, San Francisco, United States), Garmin Vivosmart HR+ (Garmin, Olathe, United States)	The tested well-known devices could be adopted to estimate steps, energy expenditure, and sleep duration with an acceptable level of accuracy in the population of interest, although clinicians should be cautious when considering other parameters for clinical and research purposes
[51]	Behavior change	Behaviors	Health	Research questions	25 interviewed users	Apple (Apple, Cupertino, United States), Xiaomi (Xiaomi, Beijing, China), Fitbit (Fitbit, San Francisco, United States), Garmin (Garmin, Olathe, United States)	Data revealed that wearables can influence users’ perceptions of self-efficacy regarding performing an activity
[52]	Health	Accuracy	Medical	Experimental	Thirty-three people with mild–moderate PD performed six two-minute indoor walks at their self-selected walking pace and at target cadences of 60, 80, 100, 120, and 140 beats/min	Fitbit Charge HR (Fitbit, San Francisco, United States), Garmin Vivosmart (Garmin, Olathe, United States)	The Garmin device was more accurate at reflecting step count across a broader range of walking cadences than the Fitbit, but neither strongly reflected intensity of activity
[15]	Behavior change	Physical activity	Health	Research questions	50 long-term wearable users based in Switzerland, used purposive sampling	Apple (Apple, Cupertino, United States), Fitbit (Fitbit, San Francisco, United States), Garmin (Garmin, Olathe, United States), Polar (Polar Electro, Kempele, Finland)	Four wearable use patterns and the associated behavior outcomes: 1) Following and compliance change, 2) Ignoring and no behavior change, 3) Combining and behavior change, and 4) Self-leading and no wearable-induced behavior change
[53]	Behavior change	Physical activity	Health	Experimental	8 wearable sensors were placed on a human subject’s body to monitor three activities: running (a1), walking (a2), and sitting (a3)	Wearable sensors	Experimental analysis of the proposed multi-level decision system found that the new method improved the accuracy and true positive rate by reducing fusion delay

Some key behavior change techniques in wearables and IoT devices as highlighted in various research articles and literature reviews were identified in [5,54]. The specific BCTs used can vary depending on the device and its intended use, whether it is for fitness tracking, chronic disease management, rehabilitation, or general health and well-being. Recent research continues to explore how these techniques can be effectively integrated into wearable and IoT technologies to maximize their impact on behavior change. For the most current and specific findings, consulting the latest peer-reviewed articles and systematic reviews in this field is advised.

**Figure 7 sensors-24-02429-f007:**
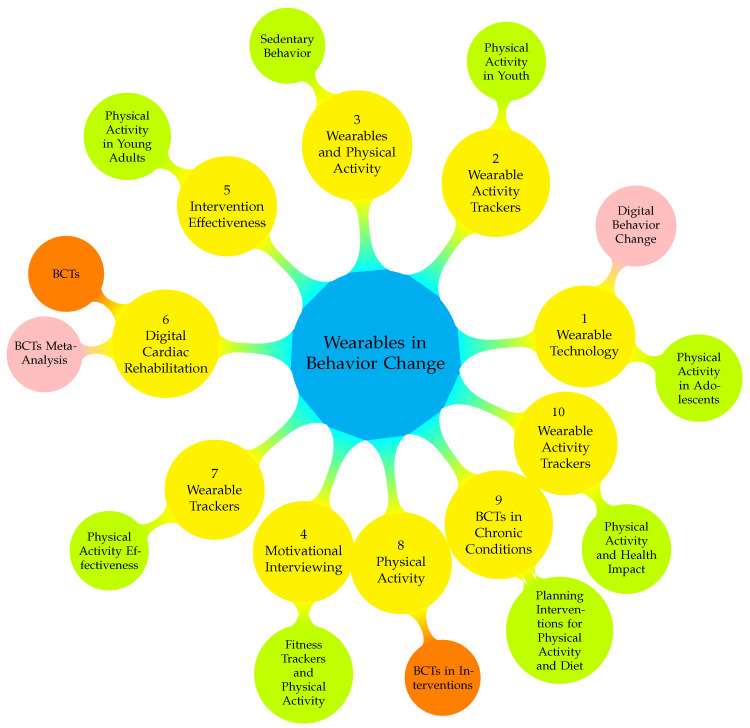
Mindmap illustrating the interconnections and thematic relationships among key research articles on the impact of wearable technologies in facilitating BCTs. 1. [55], 2. [56], 3. [57], 4. [58], 5. [59], 6. [60], 7. [61], 8. [62], 9. [63], 10. [64].

Self-Monitoring: This technique involves tracking and recording one’s own behavior, such as physical activity levels, dietary habits, or sleep patterns. Wearables and IoT devices can automate this process, making it more convenient and accurate.Feedback on Behavior: Devices often provide real-time feedback based on the data collected. This feedback can be about physical activity, heart rate, sleep quality, etc., and is used to encourage positive behavior change.Goal Setting: Many wearables and IoT devices allow users to set personal goals related to health and fitness. These goals can be tailored to the individual’s current ability and can be adjusted over time.Social Support: Integration with social networks or community platforms enables users to share their achievements and progress, fostering a sense of community and support.Rewards and Incentives: To motivate continued use and adherence to health-related behaviors, some devices incorporate reward systems such as points, badges, or sharing of achievements on social media.Reminders and Alerts: These devices can send reminders or alerts to encourage physical activity, medication adherence, or other health-related behaviors.Personalization: These devices have the ability to provide personalized information and recommendations based on the user’s behavior and preferences.Education and Information Provision: This provides users with educational content related to health, wellness, and the benefits of certain behaviors.

## 3. Methodology

Describing the process of searching for bibliographic references in the context of our research on BCTs in wearables and IoT devices, we followed a systematic and structured methodology. This approach ensured that the literature review was comprehensive, transparent, and replicable. Here is a detailed methodology of the bibliographic search:

Primary Objective: To identify and synthesize existing research on BCTs in wearables and IoT devices. Secondary Objectives: Explore specific aspects like effectiveness, user perception, technological advancements, and application areas. The search was targeted from the year 2000 to 2023 to capture the most recent and relevant studies.

Determine Search Criteria:Keywords and Phrases: we use specific and relevant terms such as “Behavior Change Techniques”, “Wearable Technology”, “IoT Devices”, and “Health Monitoring”.Inclusion and Exclusion Criteria: we define criteria based on publication year, language, type of article (e.g., peer-reviewed, conference papers, and articles), and thematic relevance.Databases and Sources: we identify suitable databases like Scopus. We also consider specialized journals and conference proceedings in the field.

Conduct the Search: Using the defined keywords, an initial search was conducted in Scopus. This search was broad to ensure the capture of a wide range of relevant literature.

The initial search results were screened based on titles and abstracts. Papers were selected based on their relevance to the topic, and we focused particularly on studies that examined the application and effectiveness of BCTs in wearables and IoT.

Selected articles underwent a full-text review for a detailed understanding of their methodologies, findings, and relevance to our research questions.

Statistical Analysis of Search Outputs: The work mentions analyzing the number of documents retrieved, average citations, and distribution of publications across fields like computer science and engineering. This helped with understanding the research landscape and its evolution over the years.

The present review references the use of figures to illustrate the paper selection process, to highlight the main keywords in the searches, and to show the number of documents per year. These visual tools assisted with better understanding and presenting the trends and patterns in the literature. Throughout the search and review process, gaps in the literature as well as trends in BCT research within wearables and IoT were identified. This helped to pinpoint areas that require further investigation. Key information from each article, such as methodologies, results, and conclusions, was extracted and summarized to support the research objectives. The quality and relevance of each article were assessed to ensure that the most reliable and pertinent information was included in the research. This process is described in Figure 8.

### 3.1. Benefits and Growth of Wearables Regarding BCTs

It is worth noting that various forms of physical activity (PA) can reduce the incidence of noncommunicable diseases, obesity, and mortality. However, rising levels of physical inactivity—as per the World Health Organization (WHO) approximately 28% of adults do not meet PA guidelines—necessitate immediate measures to augment PA levels [65]. Behavioral PA interventions, which employ cognitive and behavior techniques to modify and enhance PA behavior, have effectively boosted PA participation. Nonetheless, these interventions have typically targeted smaller groups of predominantly motivated individuals. In contrast, the integration of eHealth and mobile health solutions that encompass wearable sensors has the potential to encourage a wider population to elevate their levels of PA on a larger scale. Wearables, by monitoring specific aspects of PA through surrogate markers and providing biofeedback, hold promise for promoting PA. Notably, the WHO aims to endorse digital health concepts, and physicians can prescribe digital health solutions if their effectiveness is demonstrated [66]. Additionally, wearable-assisted interventions may prove to be more cost-effective than traditional interventions.

Furthermore, to stimulate PA, diverse BCTs can be embedded in wearables, with likely varying outcomes for promoting PA. With the constantly evolving wearable market, which introduces new models with rapidly changing features, it becomes crucial to choose the appropriate wearables that employ the right BCTs based on specific research objectives and goals related to healthy behavior [67]. Nevertheless, little is known about the distinctions among wearables and which technologies are more effective for increasing PA levels. Hence, this study seeks to explore which BCTs targeting PA behavior are integrated into commercially available high-end wearables.

The enumeration highlights several key trends and insights regarding the proliferation and impact of wearables in contemporary society. Firstly, it underscores the exponential growth of wearables over the past decade, showcasing their widespread adoption and integration into modern lifestyles, which is driven by their efficacy in health monitoring and lifestyle enhancement. Secondly, it emphasizes the crucial role of notifications in wearable usage, illustrating how these devices have become indispensable tools for staying informed and connected in an increasingly digital world. Additionally, the enumeration highlights the significance of wearables for promoting physical activity and healthy living and emphasizes their ability to motivate users and provide valuable data for improving overall well-being. Furthermore, it points out the variability in vital sign measurements among different wearable brands, emphasizing the importance of choosing devices tailored to individual needs. Lastly, it discusses the use of gamification and personalized communication by wearables to enhance self-efficacy and foster behavior change, showcasing their potential to not only inform but also inspire users towards healthier lifestyles.

Exponential Growth of Wearables: In the past decade, wearables have experienced astonishing growth in terms of adoption and popularity. This steady growth reflects how these devices have become ingrained in modern life. They are not limited to just technology enthusiasts but have also been embraced by the general population. The reason behind this phenomenon is their ability to monitor and improve people’s health and lifestyle. This trend has become a cultural phenomenon where individuals seek devices that help them achieve their personal goals more efficiently.Importance of Notifications: A survey showing that a significant number of wearable users primarily use them to receive notifications underscores the importance of connectivity and communication in our lives. These devices have become essential tools to keep us informed and connected in an increasingly digital world. Whether it is receiving important messages, social media alerts, or app updates, notifications are an integral part of people’s daily routines.Use for Physical Activity Monitoring: The observation that many people use wearables to monitor their physical activity highlights the ability of these devices to promote an active and healthy lifestyle. In addition to keeping us connected, wearables also motivate us to stay active and take care of our physical health. They provide valuable data about our activity levels, which can be a powerful tool for improving our quality of life.Variability in Vital Sign Measurement: The table showing the features of wearable devices from different brands reveals significant diversity in vital sign measurements. This underscores the need to choose a device that suits each user’s needs. Some individuals may require a more specialized device to monitor specific signs, while others may be satisfied with a more general approach.Gamification and Self-Efficacy: The use of gamification strategies and personalized communication by wearables is an indicator of their ability to not only provide information but also motivate people to adopt healthier lifestyles. Gamification creates a sense of achievement and competition that can be very effective in fostering behavior change. Furthermore, the customization of goals and communications enhances users’ self-efficacy, making them more likely to adopt and maintain healthy habits.

Figure 9 depicts the main benefits consumers want from wearable devices. Understanding how to develop optimal wearable products for consumers entails recognizing their behaviors and needs. Data analytics platforms recently examined numerous consumer interactions—blogs, forums, social media, and reviews—to pinpoint the primary advantages sought by users in wearables. Special focus was placed on under-addressed benefits to uncover potential future market prospects. These are: (1) Accuracy, a recurring demand across segments; (2) Value for Money, a topic often neglected but influential in purchasing decisions; (3) Durability, where market offerings fall short due to issues like battery life and water resistance; (4) Compactness, a universal preference for lightweight, user-friendly technology; and (5) Long-Term User Engagement, an unmet requirement stemming from companies overlooking engagement-enabling benefits. This highlights the push for insight-driven wearables to enhance user experience and drive sustained adoption. A user-centric approach aligned with technological opportunities is advocated for emotional resonance and engagement. The ever-evolving wearable tech landscape, shaped by IP, regulations, market dynamics, and consumer preferences, underpins the wearable future and delves into digital health. To delve deeper, consult *Signals Analytics*’ report on "The Future of Wearable Technology".

Table 8 presents a comprehensive classification of BCTs found in wearable activity trackers based on a thorough analysis of seven popular devices. The presence or absence of each BCT is clearly delineated, allowing for a precise assessment of the behavior strategies incorporated into these trackers. Notably, self-monitoring of activity levels, goal setting, feedback on performance, social support, reviewing past successes, and setting physical activity goals emerge as common BCTs that are consistently integrated into all assessed trackers. Conversely, several BCTs, such as fear arousal, motivational interviewing, and stress management training, are notably absent from these devices. This tabular representation offers valuable insights into the behavior change capabilities of wearable activity trackers, providing a foundation for evaluating their efficacy in promoting physical activity and informing future design enhancements to align with evidence-based techniques.

The research paper cited in [69] outlines the CALO-RE taxonomy of BCTs, which encompasses 40 distinct techniques aimed at guiding researchers and practitioners in identifying and categorizing BCTs for interventions to promote physical activity. It is important to note that this taxonomy serves the purpose of classification and identification rather than prescribing which techniques are most effective or should be adopted. The taxonomy includes techniques such as providing general and individualized information, setting behavior and outcome goals, action planning, identifying barriers and problem resolution, self-monitoring of behavior and outcomes, providing feedback on performance, instructing how to perform the behavior, and many more. Each technique is elaborated with examples to illustrate how it might be applied in practical contexts to encourage exercise and physical activity. This comprehensive taxonomy lays the foundation for designing surveys and interventions aimed at understanding participants’ inclinations toward wearables and their potential alignment with these behavior change techniques. By leveraging this taxonomy, researchers can craft targeted surveys to investigate which BCTs resonate most with wearable users and assess their effectiveness in driving behavior change in the context of physical activity promotion. The insights gained from such surveys can inform the design and optimization of wearable technologies to better align with evidence-based techniques and enhance their impact on motivating and sustaining physical activity behavior.

**Table 8 sensors-24-02429-t008:** Classification of behavior change techniques (BCTs) in wearable activity trackers [70].

Behavior Change Technique (BCT)	Presence in Trackers
Self-monitoring of activity levels	✓(All)
Goal setting	✓(6/7)
Feedback on performance	✓(All)
Social support	✓(All)
Reviewing past successes	✓(All)
Setting physical activity goals	✓(6/7)
Focusing on past and future performances	**X**
Teaching prompts and cues	**X**
Instructing on how to perform a behavior	**X**
Barrier identification or problem solving	**X**
Setting graded tasks	**X**
Prompting generalization of a target behavior	**X**
Environmental restructuring	**X**
Agreement on behavioral contract	**X**
Use of follow-up prompts	**X**
Prompt identification as a role model	**X**
Prompt anticipated regret	**X**
Fear arousal	**X**
Prompt self-talk	**X**
Prompt use of imagery	**X**
Relapse prevention or coping planning	**X**
Stress management or emotional control training	**X**
Motivational interviewing	**X**
General communication skills training	**X**

### 3.2. Search Outputs and Results

Crafting an effective search query stands as a primary hurdle to conducting a comprehensive review. To address the research inquiries at hand, the review must encompass all pertinent findings within the domain under scrutiny. Thus, the search string employed should encompass key terms pertinent to the subject matter to ensure the retrieval of pertinent results. For our literature review, articles were curated by querying the Scopus (Elsevier) databases for works on BCTs and IoT from 2000 to 2023, with selection based on relevance gleaned from titles and abstracts. The search query comprised the following terms: behavior change techniques AND (phrases in the classification presented in the Table 8).

The statistical parameters pertaining to the articles identified were as follows. We found 2728 documents on Scopus. The majority of the publications focused on the fields of computer science and engineering. The average number of citations between 2000 and 2023 for various combination of search terms was:behavior AND change AND techniques: 15,828 documents (computer science, 6338, and engineering, 12,207);behavior AND change AND techniques AND technology;behavior AND change AND techniques AND internet AND of AND things: 198 documents (computer science, 167, and engineering, 108);behavior AND change AND techniques AND wearables: 25 documents (computer science, 21, and engineering, 4).

Figure 10 illustrates the paper selection process with searches related to behavior change techniques and technology, Internet of Things, and wearables between the years 2000 and 2023. This process aims to identify the number of documents related to keywords while considering the context of the present review. Here, we can visualize a 20-year overview of how the search is narrowed down within the context of engineering and computer science to demonstrate the relationship between medical or psychological areas and engineering fields.

Figure 11 displays the top five keywords in the four main searches considered in this research. This allows us to observe the trend of primary search concepts limited to engineering and computer sciences.

Figure 12 depicts the number of documents per year for each search criterion combined with the phrase “behavior AND change AND techniques”. Each criterion refers to the classification presented in Table 8. Data are shown in Table 9. This allows us to gain insight into the trend of topics per year and the increase in citations per year, which is detailed in Figure 13. It is evident that certain keywords exhibit dominant patterns over specific years, indicating shifts in research focus and interest. Initially, the concentration was primarily on basic aspects such as “Social Support” and “Feedback on Performance”, reflecting early stages of development in this field. As years progressed—notably, from 2015 onward—there was a marked increase in the emphasis on more sophisticated elements like "Environmental Restructuring” and “Stress Management or Emotional Control Training”. This shift suggests an evolving landscape wherein the complexity of studies and applications for wearables and IoT increased. The year 2023, in particular, shows a significant spike in “Goal Setting" and “Social Support”, indicating a resurgence of interest in these foundational aspects, possibly due to new innovations or changes in societal needs. Overall, these trends highlight the dynamic nature of research in this area via adaptions to technological advancements and changing priorities. The analysis of citation trends for each of the ten specified keywords reveals distinct patterns in their academic prominence and research interest. “Social Support” leads significantly with 3594 citations, indicating its central role and consistent relevance in the field. This is followed by “Feedback on Performance” and “Goal Setting”, with 1035 and 1025 citations, respectively, underscoring their importance in both theoretical and applied research contexts. “Setting Physical Activity Goals” and “Self-monitoring of Activity Levels” also show noteworthy presences with 487 and 35 citations, respectively, reflecting a focused but substantial interest in these areas. In contrast, “Barrier Identification or Problem-Solving” and “Environmental Restructuring” have relatively fewer citations—67 and 19, respectively—suggesting these are emerging or niche areas within the field. Notably, “Prompt Use of Imagery” has yet to gain traction in the literature, as indicated by zero citations. “Stress Management or Emotional Control Training” and “Motivational Interviewing” occupy a middle ground in citation frequency with 60 and 107 citations, respectively, pointing to their recognized yet specific application in research. These trends provide valuable insights into the evolving priorities and areas of emphasis within this scholarly domain.

Figure 14 categorizes a collection of 524 research documents into four main themes related to wearable and IoT devices. ’Personalized Feedback’ (212 documents) demonstrates how these devices offer customized feedback to users by constantly monitoring activities like exercise, sleep, and diet. ‘Self-Monitoring’ (272 documents) highlights how wearables and IoT enable users to track their behaviors and progress, aiding in behavioral change. ‘Gamification Elements’ (33 documents) reveals the integration of game-like features in these devices to increase user engagement and motivation. Finally, ‘IoT and BCT Relationship’ (7 documents) explores how IoT devices enhance behavior change techniques by providing real-time data collection and personalized interactions. This figure essentially encapsulates the diverse impacts and applications of wearable and IoT technology in personal health and behavior modification.

## 4. Case Study

In the case study, the participant profile comprises 30 individuals that were carefully selected based on specific criteria to ensure a homogeneous and relevant sample for the research objectives. The age range of these participants is between 20 and 65 years, providing a broad spectrum that encompasses young adults, middle-aged individuals, and those approaching senior age, allowing for a comprehensive analysis across different stages of adulthood. Notably, all participants are non-smokers, a criterion essential for eliminating the potential confounding effects of smoking on energy consumption and vital signs. Additionally, none of the participants have a history of cardiovascular diseases, ensuring that the study’s findings are not skewed by underlying cardiac conditions, which can significantly impact both energy consumption and sleep patterns. Furthermore, the participants are not high-performance athletes, aligning the study with a more general population rather than those with exceptional physical conditioning, which could otherwise introduce significant variances in the study’s energy consumption and sleep monitoring metrics. It is also important to note that all participants have a normal weight range, as no instances of overweight conditions are reported, ensuring a more uniform baseline for analyzing the impacts of the proposed algorithm on energy consumption and sleep patterns. All participants have provided written informed consent to adhere with ethical research practices and ensure that they are fully aware of and agree with their involvement in the study. This comprehensive and detailed participant profile is crucial for the integrity and applicability of the research findings.

The research case study analyzes behavior change techniques’ impact on individuals’ key vital signs. Each behavior technique is studied individually, with participants engaging in representative activities associated with each technique and excluding the use of wearables from other techniques. Vital signs such as breathing frequency (BF), deep sleep (DS), heart rate (HR), oxygen saturation (OS), REM sleep (REMS), and temperature (T) are closely monitored throughout the study.

In commercial wearable devices, vital signs such as breathing frequency, deep sleep, heart rate, oxygen saturation, REM sleep, and temperature are measured through integrated sensors and technology designed for non-invasive and continuous monitoring. Heart rate is typically measured using optical heart rate monitors that employ light-based photoplethysmography to detect blood volume changes. Sleep stages like deep sleep and REM sleep are discerned via accelerometers and heart rate data and by using algorithms to analyze movement and physiological signals during sleep. Oxygen saturation can be gauged by some advanced wearables using pulse oximetry sensors that estimate blood oxygen levels. Breathing frequency might be tracked indirectly through movement sensors or algorithms interpreting heart rate variability. For temperature, wearables often use embedded thermistors or similar components to measure skin temperature, which, while not core body temperature, can provide valuable health insights.

In our research, a distinct group of 30 individuals was subjected to the study’s eight behavior change techniques over eight weeks with one week dedicated to each technique. Throughout the experiment, participants exclusively utilized wearables for activities corresponding to the designated technique of the week. Concurrently, vital signs were monitored to evaluate the techniques’ impact on individuals. Additionally, participants completed surveys assessing their satisfaction levels with the various activities associated with each behavior change technique. These devices offer functionalities for comfort, interaction, and validation of bodily behavior, providing real-time monitoring and feedback both to the user and other devices. The study delved into individuals’ perceptions of satisfaction with behavior techniques and averaged the impact percentage attributed to wearables on each behavior. Results indicated that techniques such as self-appraisal, goal setting, and personalization were most appreciated and yielded significant impacts facilitated by wearables.

The recorded data and basic variable analysis give us a better understanding of how BCTs affect the monitoring of vitals. The grey area on the graphics represents average adult values for each vital sign measurement. The three different measures of center that were used are mean, median, and mode [71]. For breathing frequency (BF), all used BCTs allowed for values in the normal adult range of breaths per minute. The BCT with the lowest values overall was goal setting, while the highest values were given by behavior self-monitoring and personalization. The next vital measured, deep sleep, had values that were mostly under the average adult’s percentage [72]. The BCTs that had closer results to the average were goal setting and provision of instructions. Heart rate metrics were inside the average adult’s beats per minute (bpm) range [73], yet induction was the BCT with the highest values.

BCTs are related to vital sign measurements, and basic variable analysis. Most oxygen saturation (OS) measurements are within the grey area of average adult metrics [74]. The BCT social support had values below the average adult data in all three graphs. Personalization had values slightly below the grey area, whilst goal setting and provision of instructions resulted in the highest values overall, yet the results for both of them were inside the adult average area. For REM sleep (REMS), we can note that goal setting, induction, self appraisal, and provision of instructions had values throughout all the metrics that were inside the average adult range [75]. Behavior self-monitoring and social support were the BCTs with values outside the grey area on all three data metrics. Personalization had the lowest mode value: four times less than the average adult range for REMS percentage. Almost all BCTs were similar and inside the average adult range in Celsius [76] for temperature (T) data. Behavior self-monitoring and personalization were slightly above the adult average in all graphs.

The analyses of the mean, median, and mode seen in Figure 15, Figure 16 and Figure 17 reveal intriguing insights into the behavior patterns represented. In the BF dataset, the mean and median values are fairly close, ranging around 15–16, with a slight increase towards the weekend. This suggests a moderate, somewhat uniform level of behavior across the week with a minor peak on weekends. The mode values, however, show more variation, indicating specific frequent behaviors on certain days. The DS dataset has lower mean and median values, averaging around 12–13. This consistency suggests a steady behavior pattern, though the mode values again vary, pointing towards certain predominant behaviors on different days. HR’s data exhibit significantly higher mean and median values, around 80, suggesting more intense behavior patterns in this domain. The closeness of these measures indicates a consistent level of behavior, yet the mode suggests variations in the most common behaviors. In OS, the mean and median are remarkably stable across the week, hovering around 94–95, denoting a very consistent behavior pattern. The mode values show minor fluctuations, which might indicate specific recurring behaviors. REMS data show mean and median values in the higher teens, with slight increases on weekends. This implies a somewhat consistent pattern with minor upticks during weekends. The modes, which vary around these values, highlight specific prevalent behaviors. T’s dataset shows extremely consistent mean and median values, around 37.2–37.3, indicating a very stable behavior pattern across the week. The mode values, closely aligned with the mean and median, reinforce this consistency. Overall, these statistics paint a picture of varying behavioral intensities and consistencies across different domains and days, with some showing uniform behavior throughout the week while others exhibit specific prevalent behaviors on certain days.

For BF, all BCTs had results inside the average adult area; yet the BCTs with the most stable data were behavior self-monitoring and personalization. DS had better results with goal setting and provision of instructions as BCTs. All BCTs performed well, inside the average grey area, for HR. The most stable BCTs for HR were provision of instructions and behavior self-monitoring. OS had induction and gamification as the steadiest BCTs, and goal setting had the highest values. Goal setting, self-appraisal, induction, and provision of instructions were the best BCTs for REMS. For T, induction, provision of instructions, self-appraisal, and social support obtained the most stable results within the established grey area. The BCTs with the most effectiveness overall according to the analysis above are: provision of instructions, goal setting, and induction.

Table 10, Table 11, Table 12, Table 13, Table 14 and Table 15 reveal a wide variance in values for each day of the week. Notably, the HR domain exhibits significantly higher variances compared to others, suggesting greater fluctuations in behavioral patterns within this category. The T domain, in contrast, presents relatively low variance, indicating more consistent behavior. Across all domains, there is no clear trend in variance linked to specific days of the week, signifying that the impact of days on behavioral change techniques is not uniform. This diverse range in variances underscores the complexity of human behavior and the challenges in predicting or influencing it through various techniques. Moreover, these findings suggest that the effectiveness of BCTs may vary considerably depending on the day and the specific domain in which they are applied, emphasizing the need for tailored approaches in behavior change interventions.

## 5. Discussion

The current research literature highlights the potential of wearable devices to positively impact behavior change techniques and influence various aspects of people’s health. Integrating wearables with behavior change interventions such as real-time feedback, personalized recommendations, and stress-reduction techniques has shown promising results in improving physical activity levels, sleep quality, oxygen saturation, and stress management. These changes in behavior are often reflected in vital signs, including heart rate, deep sleep metrics, oxygen saturation levels, and body temperature. As wearable technologies continue to advance, their role in behavior change interventions and health promotion is expected to grow, leading to better health outcomes for individuals.

Leveraging wearables and IoT devices for behavior change represents a profound shift in health and behavior management. These technologies, through real-time data capture and analysis, offer a personalized window into individual habits and health metrics, fundamentally transforming the way individuals engage with their health and wellness journeys. By integrating behavior change techniques, these devices not only provide a platform for continuous self-monitoring but also create an interactive, engaging experience that fosters sustained behavioral change. This is more than just a technological advancement; it is a paradigm shift in health management, where individuals are empowered with insights and feedback that make health management proactive, personalized, and responsive. The analytical depth of these technologies lies in their ability to process complex health data, provide contextual feedback, and motivate change through tailored interventions. This convergence of technology and human behavior highlights a future where personalized health management is seamlessly integrated into everyday life, promising significant improvements in health outcomes and overall well-being.

### 5.1. Emerging Solutions

In the realm of emerging solutions, information technologies have catalyzed groundbreaking advancements in promoting health and well-being, particularly through the proliferation of wearables and Internet of Things devices [77]. These innovative technologies are embedded with sensors and sophisticated data processing capabilities and hold immense potential to revolutionize behavior change interventions by offering personalized feedback and insights [78]. By harnessing behavior change techniques, wearables and IoT devices empower individuals to adopt healthier lifestyles and achieve sustainable behavior modifications. Through a meticulous examination of contemporary research studies, this work aims to unravel the intricate mechanisms and strategies employed by these technologies to facilitate behavior change, shedding light on their transformative impact on individuals and society [79].

One of the key strengths of wearables and IoT devices lies in their ability to provide personalized feedback and recommendations tailored to individual needs and preferences. By continuously monitoring and analyzing user data, these devices offer valuable insights into daily activities, sleep patterns, and dietary habits, enabling users to make informed decisions and strive towards their health goals [80]. Moreover, the incorporation of gamification elements and social connectivity features enhances user engagement and motivation, fostering a sense of community and accountability [81]. As we delve deeper into this dynamic field, it becomes increasingly evident that wearables and IoT devices are not just technological innovations but are powerful catalysts for positive behavior change that are poised to reshape the landscape of health and well-being in profound ways.

### 5.2. Limitations

In the field of limitations, despite the promising potential of behavior change techniques facilitated by information technologies and wearables, several challenges and constraints warrant attention. One significant limitation lies in the reliance on user engagement and adherence to the prescribed interventions. While wearables and IoT devices offer personalized feedback and gamified features to enhance motivation, sustaining user interest and commitment over time remains a formidable obstacle. Studies have highlighted instances of user disengagement and abandonment of wearable devices, indicating the need for innovative strategies to address these issues. Additionally, concerns regarding data privacy and security pose another limitation to the widespread adoption and effectiveness of these technologies. As wearables collect and transmit sensitive health-related data, ensuring robust privacy safeguards and regulatory compliance becomes imperative to mitigate potential risks and maintain user trust.

Furthermore, the effectiveness of behavior change interventions facilitated by wearables and IoT devices may vary across diverse populations and contexts, presenting a challenge to their generalizability and scalability. Factors such as socioeconomic status, cultural background, and technological literacy can influence individuals’ receptivity and response to these interventions. Addressing these disparities and tailoring interventions to meet the specific needs and preferences of different demographic groups is essential to maximize their impact and reach. Additionally, the dynamic nature of technology evolution and obsolescence introduces uncertainties regarding the long-term sustainability and compatibility of wearable devices and IoT platforms. As newer models and technologies emerge, ensuring seamless integration and interoperability with existing infrastructures and protocols becomes crucial to prevent fragmentation and to ensure continuity of care and support for behavior change initiatives. In navigating these limitations, interdisciplinary collaboration and iterative refinement of intervention designs are essential to foster innovation and enhance the effectiveness and inclusivity of the behavior change strategies enabled by information technologies and wearables.

## 6. Conclusions

In conclusion, this research paper sheds light on the significant contribution of information technologies, particularly wearables and Internet of Things devices, in fostering positive behavior change and promoting individuals’ well-being. By exploring the effectiveness and impact of behavior change techniques facilitated by these technologies, the study underscores their potential to empower individuals to adopt healthier lifestyles and achieve long-term behavior modifications. Through an examination of relevant studies, the paper highlights the importance of precise definitions and thorough reporting of BCTs, as well as the establishment of a coherent connection between interventions and theoretical mechanisms. This comprehensive understanding is crucial for designing and implementing effective interventions for behavior change using wearable and IoT technologies.

This comprehensive review underscores the significant advancements and potential of behavior change techniques in wearables and IoT devices, particularly within the realms of engineering and computer science. However, to further strengthen the impact and utility of these technologies, it is imperative that future research endeavors focus on developing clear recommendations for their application. Such recommendations should not only include guidelines for effective integration of BCTs in wearables and IoT systems but also elaborate on how these technologies can be tailored to address diverse user needs and contexts. Moreover, future studies should explore the practical implications of these technologies in real-world scenarios: assessing their effectiveness in various settings such as healthcare, sports rehabilitation, and everyday wellness monitoring. This approach will not only contribute to a more profound understanding of the interplay between technology, human behavior, and health but also pave the way for innovative solutions that resonate with the evolving demands of our society. As such, our manuscript provides a foundational platform for further exploration and innovation in the dynamic intersection of engineering, computer science, and health technologies, encouraging researchers and practitioners to delve deeper into this promising field.

Moreover, the research emphasizes the key advantages of wearables and IoT devices for promoting behavior change, including their ability to provide personalized feedback, facilitate self-monitoring, incorporate gamification elements, and leverage social influence. These features make the behavior change process more enjoyable, increase individuals’ adherence to their goals, and foster a sense of community and accountability. The integration of behavior change techniques with wearables has opened up new avenues for personalized health interventions and self-management strategies, ultimately leading to improvements in individuals’ health behaviors and outcomes.

The comprehensive analysis of BCTs across various vital signs, with a focus on mean, median, and mode, underscores the nuanced effectiveness of these techniques in behavior modulation. While each BCT shows efficacy within specific parameters, the variability in results across different days of the week highlights the dynamic nature of human behavior and the complexity of influencing it through systematic interventions. This variability necessitates a tailored approach to behavior change interventions that takes into account individual differences and day-to-day fluctuations. Moreover, the identification of certain BCTs as particularly effective for stabilizing or improving specific vital signs opens avenues for targeted, personalized health strategies. This insight can significantly contribute to research and development in the science of behavior change to allow for more precise and effective application of BCTs in health interventions, potentially leading to improved health outcomes and more efficient resource utilization in healthcare and wellness programs. The implications of this study extend beyond the scope of individual health by offering valuable information for the broader field of behavioral science, where understanding and influencing human behavior is central.

Looking ahead, as technology continues to advance, it is essential to further explore and understand the effectiveness and long-term impact of these behavior change techniques fostered by wearables and IoT devices. By harnessing the full potential of these technologies and continuing to innovate in the field of behavior change interventions, we can strive towards improving individuals’ health and quality of life on a broader scale.

## Figures and Tables

**Figure 1 sensors-24-02429-f001:**
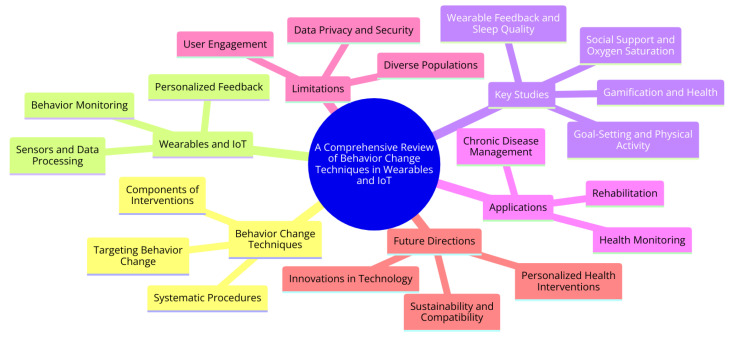
Mindmap diagram of the research.

**Figure 2 sensors-24-02429-f002:**
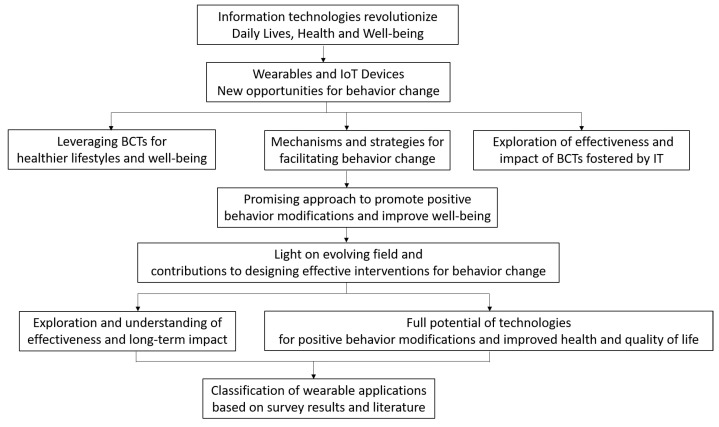
Main points and progression of the research.

**Figure 3 sensors-24-02429-f003:**
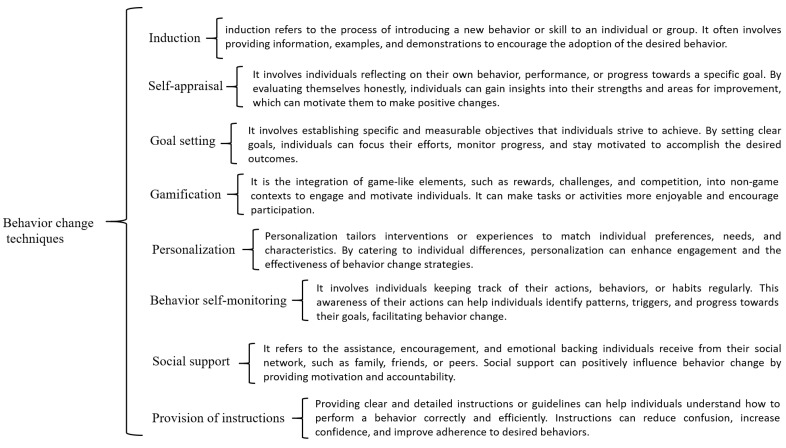
Behavior change technique definition scheme.

**Figure 4 sensors-24-02429-f004:**
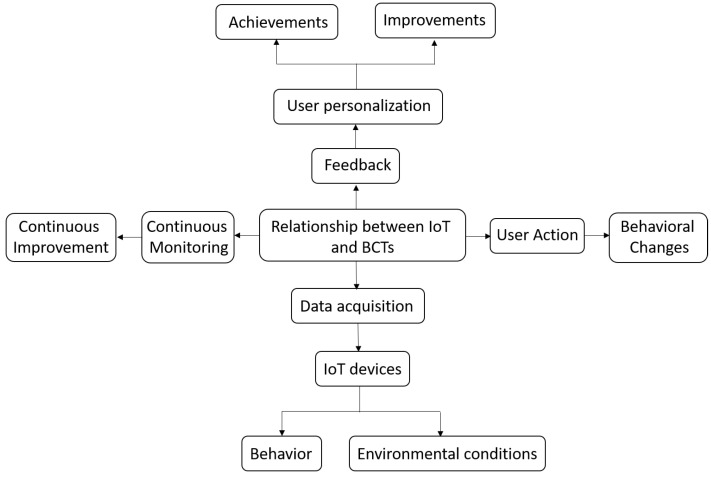
IoT and BCT relationship diagram.

**Figure 5 sensors-24-02429-f005:**
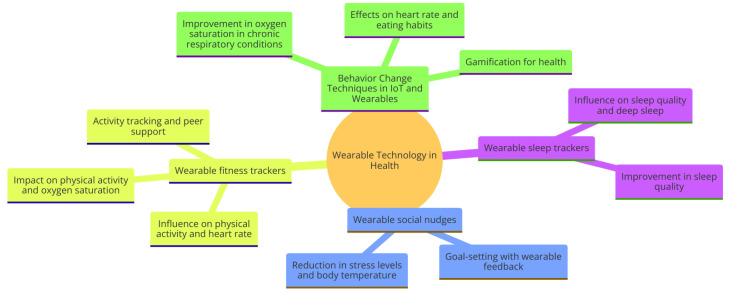
Related works analysis summary for behavior change techniques and wearables.

**Figure 6 sensors-24-02429-f006:**
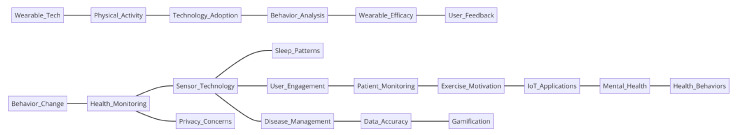
Relationships among the main citations of the related works.

**Figure 8 sensors-24-02429-f008:**
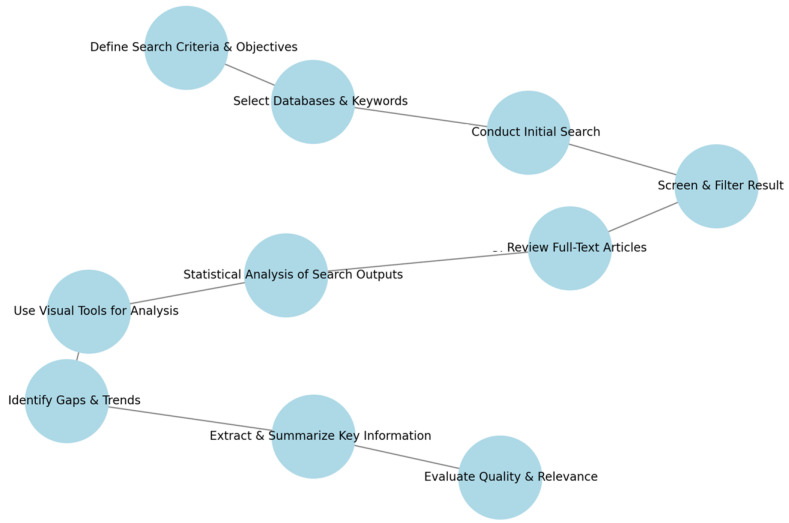
Research methodology—search process.

**Figure 9 sensors-24-02429-f009:**
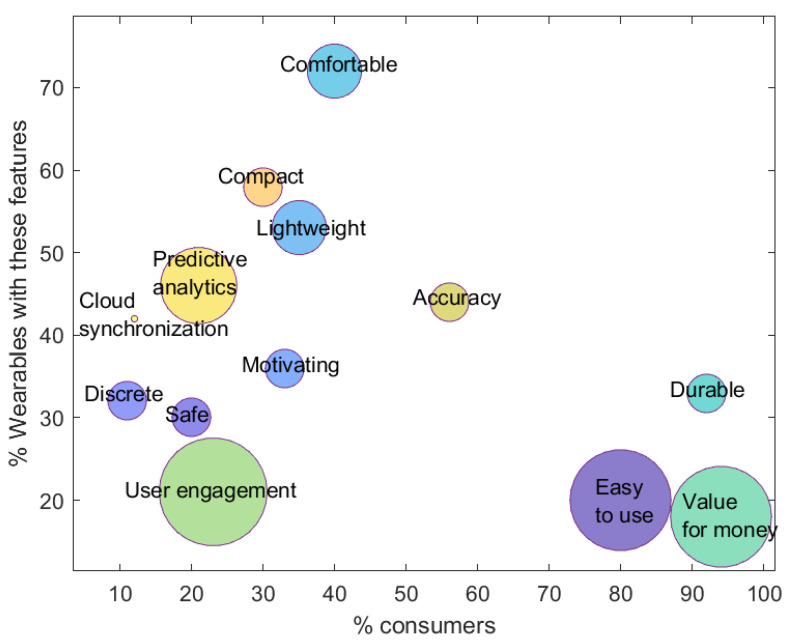
Benefits consumers want their wearable devices to have [68].

**Figure 10 sensors-24-02429-f010:**
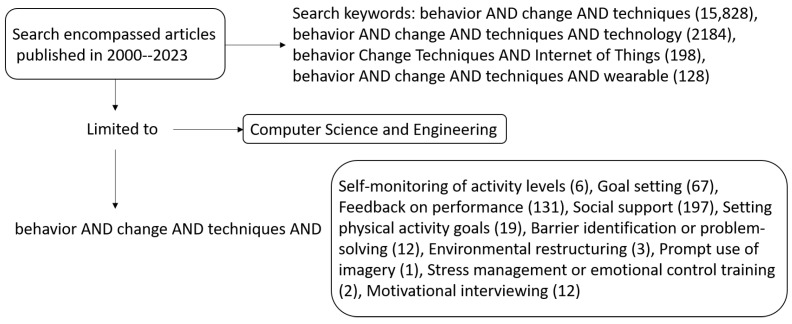
Paper selection process using Scopus for publications from 2000 to 2023.

**Figure 11 sensors-24-02429-f011:**
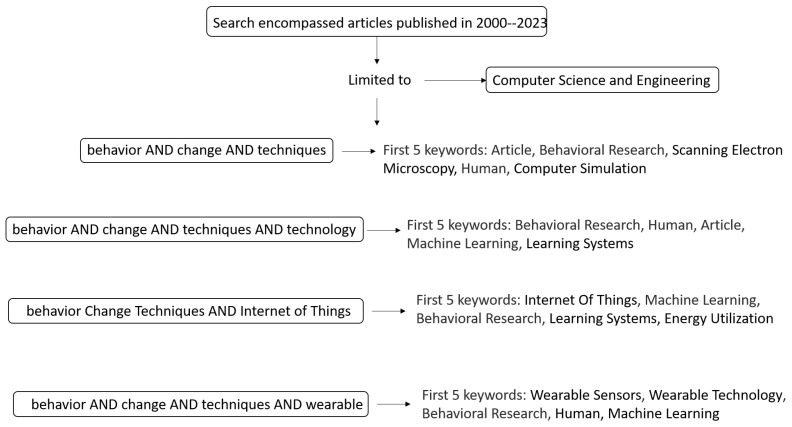
First keywords by search process between 2000 and 2023 according to Scopus.

**Figure 12 sensors-24-02429-f012:**
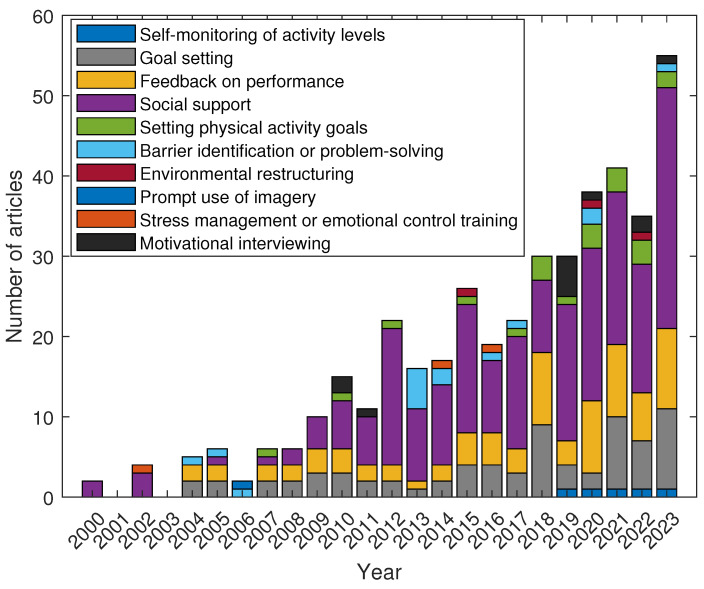
Number of research papers on behavior change techniques published between 2000 and 2023 according to Scopus.

**Figure 13 sensors-24-02429-f013:**
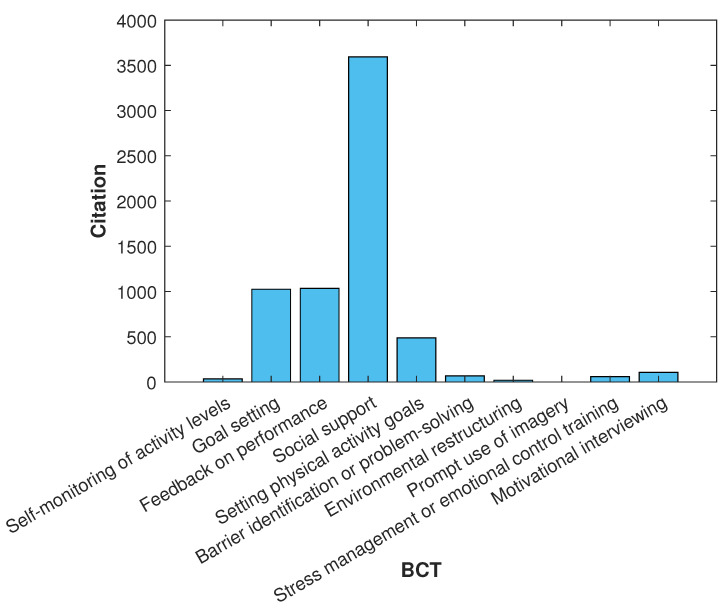
Citations on behavior change techniques published between 2000 and 2023 according to Scopus.

**Figure 14 sensors-24-02429-f014:**
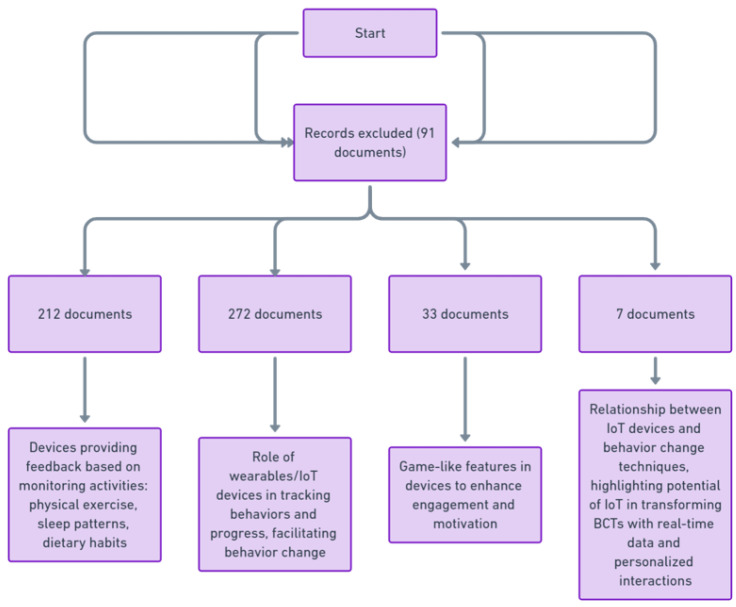
Exploring the multifaceted impact of wearable and IoT devices.

**Figure 15 sensors-24-02429-f015:**
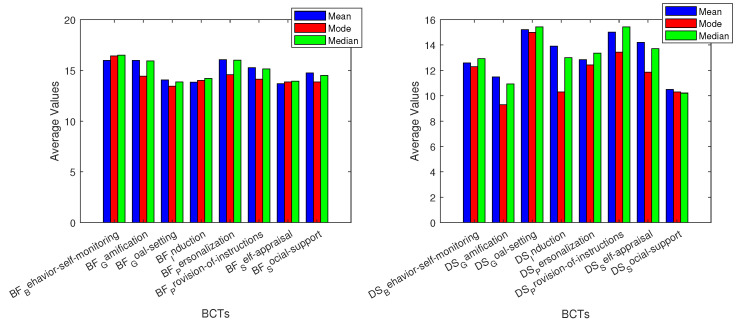
Mean, median, and mode values for BF and DS.

**Figure 16 sensors-24-02429-f016:**
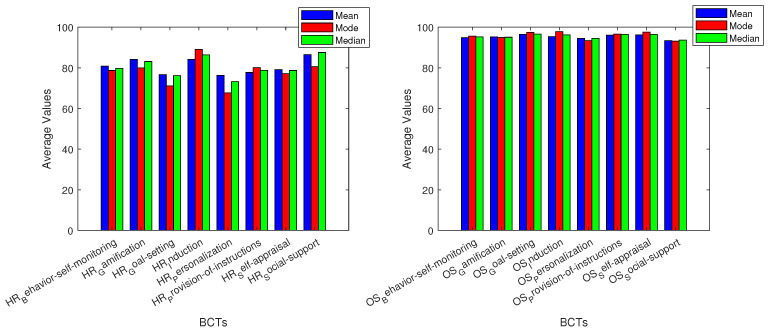
Mean, median, and mode values for HR and OS.

**Figure 17 sensors-24-02429-f017:**
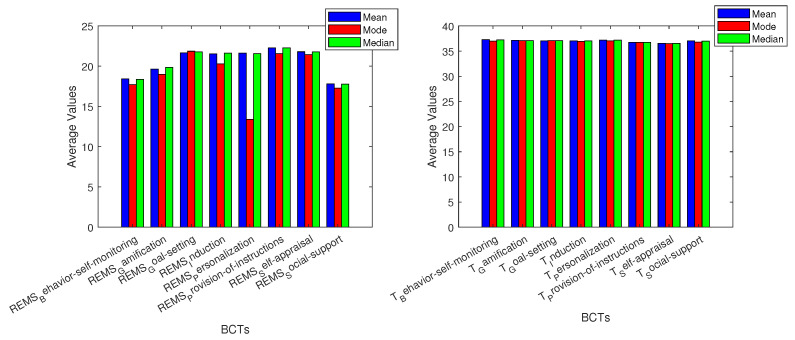
Mean, median, and mode values for REMS and T.

**Table 1 sensors-24-02429-t001:** Behavior change technique (BCT) main categories.

Environmental Modification	Goal Setting and Planning	Information and Education	Self-Regulation and Coping Strategies
Adding to the physical environment to encourage or facilitate the target behavior.	Setting specific, achievable goals related to the target behavior.	Providing information on consequences of behavior in general.	Creating awareness of the difference between current behavior and desired goals.
Modifying the environment to reduce barriers to the target behavior.	Setting specific, achievable goals related to the desired outcomes of the behavior change.	Providing personalized information regarding consequences of behavior to the individual.	Planning strategies to prevent relapse and cope with potential challenges.
Involves altering the physical environment to encourage or facilitate the target behavior.	Developing specific plans and strategies for implementing the target behavior.	Involves providing information about behavior consequences to raise awareness and promote behavior change.	Focuses on creating awareness of behavior–goal gaps and developing plans to prevent relapse and address challenges.

**Table 2 sensors-24-02429-t002:** Behavior change technique (BCT) main categories.

Self-Monitoring and Feedback	Skill Training and Demonstration	Social Reinforcement and Incentives	Social Support and Comparison
Tracking and recording the target behavior to increase awareness and promote behavior change.	Providing clear instructions on how to perform the target behavior.	Providing social rewards or praise for achieving the behavior change goals.	Receiving emotional support and encouragement from others to promote behavior change.
Showing a demonstration of the target behavior to facilitate learning.	Creating incentives based on social aspects or interactions.	Encouraging behavior change by comparing one’s actions or progress to others’.	
Providing information or feedback on the target behavior to increase motivation and awareness.	Involves providing clear instructions and demonstrations to facilitate the performance of the target behavior.	Providing social rewards and praise for achieving behavior change goals, often involving celebrating milestones with friends or family.	Involves receiving emotional encouragement from others to foster behavior change, often through friends or family, providing motivation for maintaining a routine.

**Table 3 sensors-24-02429-t003:** Related works on wearables and behavior change techniques.

Reference	Behavior Change Technique	Study Design	Comparison Group	Study Duration
[26]	Self-monitoring using wearables	2-arm pilot	Intervention group	8 weeks
[27]	Gamification for physical activity	3-arm randomized controlled trial	Control group	12 weeks
[28]	Wearable feedback for sedentary behavior	Single-arm pre–post	Not Applied	4 weeks
[29]	Social support through wearables	2-arm repeated-measure experimental	Comparison group	16 weeks
[21]	Wearable prompts for healthy eating	2-arm quasi-experimental	Control group	6 months
[30]	Goal setting with wearable feedback	2-arm quasi-experimental	Control group	12 weeks
[31]	Wearable social nudges for physical activity	2-arm pilot	Intervention group	6 weeks
[32]	Sleep improvement through wearables	Single-arm pre–post	Not Applied	4 weeks
[33]	Gamification for healthy eating	3-arm randomized controlled trial	Control group	8 weeks
[34]	Activity tracking and peer support	2-arm repeated-measure experimental	Comparison group	16 weeks

**Table 4 sensors-24-02429-t004:** Overview of wearable devices by category, including head, limb, and torso, with specific examples and use cases.

Head-Wearable Devices	Limb-Wearable Devices	Torso-Wearable Devices
Glasses, helmets, headbands, etc.	Smart watches, bracelets, etc.	Suits, belts, underwear, etc.
Virtual reality, augmented reality for telemedicine	Monitoring physiological parameters	Electronic products in fabrics
Application in medical education, intraoperative navigation	Lower-limb wearables for rehabilitation	

**Table 5 sensors-24-02429-t005:** Classification of wearable device applications in health and safety monitoring, chronic disease management, and disease diagnosis and treatment.

Health and Safety Monitoring	Chronic Disease Management	Diagnosis and Treatment of Diseases
Monitors gait, posture, vital signs in real time	Changes passive disease treatment to active monitoring	Early detection of Alzheimer’s through gait
Supports older adults, children, pregnant women, patient groups	Cardiovascular diseases, pulmonary diseases, diabetes management	Monitoring respiratory diseases, cardiac anomalies, urinary diseases
Use in disease diagnosis, treatment, and rehabilitation	Hypertension, urinary diseases	Cognitive rehabilitation, aids for disabilities
Therapeutic applications in early stages		

**Table 6 sensors-24-02429-t006:** Roles of wearable devices in sports and cognitive rehabilitation, highlighting their applications and benefits in each area.

Sports Rehabilitation	Cognitive Rehabilitation
Focuses on stroke, brain trauma, spinal cord injury	Utilizes VR technology for cognitive impairment
Monitors gait parameters, guides exercises	Provides immersive experiences, memory recovery
Supports limb hemiplegia recovery, upper limb training	VR-based wearable devices for cognitive dysfunction
Improves awareness, memory, and function recovery	
Assists people with disabilities through smart devices	

**Table 9 sensors-24-02429-t009:** Data per year from Figure 12.

Year	Self-Monitoring of Activity Levels	Goal Setting	Feedback on Performance	Social Support	Setting Physical Activity Goals	Barrier Identification or Problem-Solving	Environmental Restructuring	Prompt Use of Imagery	Stress Management or Emotional Control Training	Motivational Interviewing
2000	0	0	0	2	0	0	0	0	0	0
2001	0	0	0	0	0	0	0	0	0	0
2002	0	0	0	3	0	0	0	1	0	0
2003	0	0	0	0	0	0	0	0	0	0
2004	0	2	2	0	0	1	0	0	0	0
2005	0	2	2	1	0	1	0	0	0	0
2006	0	0	0	0	0	1	0	1	0	0
2007	0	2	2	1	1	0	0	0	0	0
2008	0	2	2	2	0	0	0	0	0	0
2009	0	3	3	4	0	0	0	0	0	0
2010	0	3	3	6	1	0	0	0	2	0
2011	0	2	2	6	0	0	0	0	1	0
2012	0	2	2	17	1	0	0	0	0	0
2013	0	1	1	9	0	5	0	0	0	0
2014	0	2	2	10	0	2	0	1	0	0
2015	0	4	4	16	1	0	1	0	0	0
2016	0	4	4	9	0	1	0	1	0	0
2017	0	3	3	14	1	1	0	0	0	0
2018	0	9	9	9	3	0	0	0	0	0
2019	1	3	3	17	1	0	0	0	0	5
2020	1	2	9	19	3	2	1	0	0	1
2021	1	9	9	19	3	0	0	0	0	0
2022	1	6	6	16	3	0	1	0	0	2
2023	1	10	10	30	2	1	0	0	1	0

**Table 10 sensors-24-02429-t010:** Variance for each day of the week across different BCTs for BF.

BCT	Monday	Tuesday	Wednesday	Thursday	Friday	Saturday	Sunday
BF_Behavior-self-monitoring	14.53	12.83	15.27	14.05	16.86	11.36	10.92
BF_Gamification	7.01	9.54	12.95	17.44	10.96	16.40	12.40
BF_Goal-setting	14.02	16.14	11.61	10.20	13.68	7.93	10.17
BF_Induction	12.30	14.81	9.79	9.55	19.15	14.11	13.50
BF_Personalization	12.32	12.81	14.17	11.20	17.24	11.72	13.83
BF_Provision-of-instructions	11.72	12.74	8.25	11.56	15.50	14.64	12.74
BF_Self-appraisal	14.60	9.87	14.65	10.63	11.66	8.10	5.03
BF_Social-support	25.43	18.37	21.10	9.20	29.83	16.99	23.34

**Table 11 sensors-24-02429-t011:** Variance for each day of the week across different BCTs for DS.

BCT	Monday	Tuesday	Wednesday	Thursday	Friday	Saturday	Sunday
DS_Behavior-self-monitoring	22.25	20.60	24.12	35.47	18.05	33.48	29.01
DS_Gamification	27.76	17.41	19.32	24.48	23.73	21.76	23.39
DS_Goal-setting	19.48	18.67	16.11	18.86	16.16	13.84	24.63
e DS_Induction	19.96	21.91	24.23	18.05	29.49	25.75	27.18
DS_Personalization	13.79	20.19	19.31	22.83	23.68	37.08	27.29
DS_Provision-of-instructions	14.52	15.14	24.38	12.28	17.69	23.10	13.54
DS_Self-appraisal	22.84	30.84	29.54	17.40	20.87	26.51	16.78
DS_Social-support	26.38	14.96	15.64	12.23	21.48	24.77	23.83

**Table 12 sensors-24-02429-t012:** Variance for each day of the week across different BCTs for HR.

BCT	Monday	Tuesday	Wednesday	Thursday	Friday	Saturday	Sunday
HR_Behavior-self-monitoring	103.84	89.44	106.16	105.77	100.67	108.62	95.76
HR_Gamification	55.44	70.10	65.02	58.60	68.45	66.02	47.03
HR_Goal-setting	90.34	89.01	77.82	80.10	85.21	84.29	79.61
HR_Induction	118.72	115.96	105.03	121.90	110.56	107.52	105.66
HR_Personalization	104.46	100.32	108.99	107.90	121.14	116.23	111.38
HR_Provision-of-instructions	63.84	59.16	60.11	59.99	53.69	62.78	58.05
HR_Self-appraisal	57.15	64.56	57.55	54.28	57.34	52.86	49.68
HR_Social-support	91.75	90.52	95.97	78.15	89.24	101.15	100.16

**Table 13 sensors-24-02429-t013:** Variance for each day of the week across different BCTs for OS.

BCT	Monday	Tuesday	Wednesday	Thursday	Friday	Saturday	Sunday
OS_Behavior-self-monitoring	16.42	14.26	8.55	12.89	7.14	13.87	8.74
OS_Gamification	9.40	12.71	9.98	11.36	6.37	9.55	11.43
OS_Goal-setting	8.55	7.57	12.37	7.08	13.34	6.70	9.70
OS_Induction	11.09	10.81	14.53	12.91	291.70	9.84	7.70
OS_Personalization	12.21	12.33	10.40	11.00	13.36	12.74	11.91
OS_Provision-of-instructions	13.00	13.90	10.44	10.37	10.17	12.82	9.44
OS_Self-appraisal	10.03	12.64	12.70	6.05	8.19	7.29	9.59
OS_Social-support	12.53	9.96	11.76	9.52	10.06	11.01	11.44

**Table 14 sensors-24-02429-t014:** Variance for each day of the week across different BCTs for REMS.

BCT	Monday	Tuesday	Wednesday	Thursday	Friday	Saturday	Sunday
REMS_Behavior-self-monitoring	11.73	16.05	15.73	11.52	16.98	14.25	17.82
REMS_Gamification	21.61	20.79	14.56	17.32	15.79	18.41	10.10
REMS_Goal-setting	11.42	8.34	9.58	14.38	16.99	14.13	13.93
REMS_Induction	11.57	9.90	10.03	13.08	16.30	18.74	10.55
REMS_Personalization	12.88	15.25	11.44	11.26	19.68	12.62	10.57
REMS_Provision-of-instructions	10.59	10.10	21.80	6.64	12.74	6.95	13.02
REMS_Self-appraisal	13.96	15.60	9.56	7.95	6.37	7.77	16.40
REMS_Social-support	18.64	10.31	11.91	9.73	11.21	11.50	7.07

**Table 15 sensors-24-02429-t015:** Variance for each day of the week across different BCTs for T.

BCT	Monday	Tuesday	Wednesday	Thursday	Friday	Saturday	Sunday
T_Behavior-self-monitoring	0.16	0.11	0.11	0.19	0.14	0.10	0.14
T_Gamification	0.30	0.15	0.13	0.20	0.11	0.13	0.11
T_Goal-setting	0.15	0.15	0.12	0.12	0.21	0.10	0.22
T_Induction	0.26	0.14	0.19	0.17	0.12	0.09	0.18
T_Personalization	0.22	0.18	0.19	0.18	0.09	0.20	0.15
T_Provision-of-instructions	0.24	0.15	0.08	0.11	0.15	0.17	0.12
T_Self-appraisal	0.14	0.23	0.16	0.18	0.22	0.11	0.19
T_Social-support	0.09	0.16	0.15	0.17	0.17	0.19	0.16

## Data Availability

Data are contained within the article.

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
