# Peer review of "A Comprehensive Review of Behavior Change Techniques in Wearables and IoT: Implications for Health and Well-Being"

_sensors, 2024, doi:10.3390/s24082429_

Round 1

Reviewer 1 Report

Comments and Suggestions for Authors

1- The paper briefs a lot of information about Iot and Wearables systems with Behavior Change Techniques. There are a lot of titles and a lot of texts, and they have reduced the readability of the work.

2- The manuscript must contain a section or paragraph about the organization of the paper 

3- The contributions do not clearly define the difference between this survey when compared with similar works. 

4- There are not enough figures and flowcharts to describe the aim of the survey.

5- The visual quality of the images is not enough to understand the case.

6- The difference in each method must be tabulated with more technical and numerical results. Also, all papers are discussed with only a paragraph. It must be given some information and visual supplementary figures.

 7- There is no graphical representation of the roadmap of the section. 

8- It is not enough to summarize the literature studies alone. Authors must focus on the architectures or figures of the proposed models of some literature studies.

Author Response

Dear

Editor

Sensors

We are submitting the paper:

“A Comprehensive Review of Behavior Change Techniques in Wearables and IoT: Implications for Health and Well-being”

Authored by: Carolina Del-Valle-Soto*, Juan Carlos López-Pimentel, Javier Vázquez-Castillo, Juan Arturo Nolazco-Flores, Ramiro Velázquez, José Varela-Aldás and Paolo Visconti.

We would like to thank the reviewers and editors for their detailed analysis of the manuscript; the comments are very valuable to us. In the revised version of the paper, we have incorporated the all changes recommended by the reviewers.

Comments to all observations and suggestions including point-by-point responses are addressed in the following text.

Reviewer 1 comments

Comment 1: The paper briefs a lot of information about Iot and Wearables systems with Behavior Change Techniques. There are a lot of titles and a lot of texts, and they have reduced the readability of the work.

Response: Many thanks to the Reviewer for his/her invaluable interest in the comments on this manuscript. We have removed several subtitles to make the text more readable, and we have reorganized the position of some paragraphs.

Comment 2: The manuscript must contain a section or paragraph about the organization of the paper.

Response: We have added a text at the end of the introduction presenting all sections of the manuscript. Thank you for pointing that out.

The manuscript is organized into sections distributed as follows. The Introduction describes the main BCTs and IoT and their applications in the industry. The Related Work section represents the impact of BCTs on design studies and compares current literature. The Methodology section shows the search for publications, keywords, and citations on works related to computer science and engineering. The next section is a Case Study in which we contribute to the state of the art with an experimental case of applying BCTs to wearables. Subsequently, we present the Discussion section. Finally, we explain the Conclusion section.

Comment 3: The contributions do not clearly define the difference between this survey when compared with similar works.

Response: Many thanks. We have added a clearer paragraph highlighting the specific contribution of the paper in the Motivation subsection.

This research work makes a significant contribution to the existing literature by addressing a notable gap in the field. Specifically, the scarcity of literature reviews pertaining to wearables and behavior change techniques underscores the novelty and importance of this study. Furthermore, by introducing a new case study to the current body of literature, this review adds valuable insights and empirical evidence to further enrich our understanding of the subject matter. Through meticulous analysis and synthesis of relevant literature, this research not only consolidates existing knowledge but also paves the way for future investigations and advancements in this emerging field. The primary objective of the research paper is to provide a comprehensive review of BCTs as applied within the realm of IoT devices, especially wearables, in the fields of engineering and computer science. This study addresses the increasing integration of technology into daily life and aims to understand the impact of BCTs on user behavior, particularly focusing on their efficacy and implications when implemented in wearable devices. By exploring a case study on the application of BCTs in monitoring vital signs using wearables, the research highlights the profound psychological and social implications associated with technology adoption. It seeks to illuminate how BCTs in wearables can influence user behavior, either positively or negatively. The research makes a significant contribution by filling a notable gap in the literature, offering new insights and empirical evidence, thereby enriching the understanding of the crucial intersection of technology and human behavior.

Comment 4: There are not enough figures and flowcharts to describe the aim of the survey.

Response: We have introduced further mental frameworks regarding the key points of our research.

The scheme represented in Figure 1 describes the main points and progression of the research, starting from the revolutionization of daily lives by information technologies, through the introduction of wearables and IoT devices, leveraging behavior change techniques, exploring effectiveness and impact, understanding mechanisms, shedding light on the evolving field, and finally, harnessing the full potential of technologies for positive behavior modifications and improved health and quality of life. Additionally, it includes the classification of wearable applications based on survey results and literature.

Comment 5: The visual quality of the images is not enough to understand the case.

Response: The reviewer is absolutely correct, and we have substantially altered the presentation and analysis of the results to ensure they are truly understandable and contribute meaningfully to the literature on the impact of wearable use in BCTs.

The analysis of the mean, median, and mode seen in Figures 9,10, and 11 reveal intriguing insights into the behavior patterns represented. In the BF dataset, the mean and median values are fairly close, ranging around 15-16, with a slight increase towards the weekend. This suggests a moderate, somewhat uniform level of behavior across the week with a minor peak on weekends. The mode values, however, show more variation, indicating specific frequent behaviors on certain days. The DS dataset has lower mean and median values, averaging around 12-13. This consistency suggests a steady behavior pattern, though the mode values again vary, pointing towards certain predominant behaviors on different days. HR's data exhibit significantly higher mean and median values, around 80, suggesting more intense behavior patterns in this domain. The closeness of these measures indicates a consistent level of behavior, yet the mode suggests variations in the most common behaviors. In OS, the mean and median are remarkably stable across the week, hovering around 94-95, denoting a very consistent behavior pattern. The mode values show minor fluctuations, which might indicate specific recurring behaviors. REMS data show mean and median values in the higher teens, with slight increases on weekends. This implies a somewhat consistent pattern with minor upticks during weekends. The modes, varying around these values, highlight specific prevalent behaviors. T's dataset shows extremely consistent mean and median values, around 37.2-37.3, indicating a very stable behavior pattern across the week. The mode values, closely aligned with the mean and median, reinforce this consistency. Overall, these statistics paint a picture of varying behavioral intensities and consistencies across different domains and days, with some showing uniform behavior throughout the week while others exhibit specific prevalent behaviors on certain days.

Tables 9 to 14 reveal a wide variance in values for each day of the week. Notably, the HR domain exhibits significantly higher variances compared to others, suggesting greater fluctuations in behavioral patterns within this category. The T domain, in contrast, presents relatively low variance, indicating more consistent behavior. Across all domains, there is no clear trend in variance linked to specific days of the week, signifying that the impact of days on behavioral change techniques is not uniform. This diverse range in variances underscores the complexity of human behavior and the challenges in predicting or influencing it through various techniques. Moreover, these findings suggest that the effectiveness of BCTs may vary considerably depending on the day and the specific domain in which they are applied, emphasizing the need for tailored approaches in behavior change interventions.

Comment 6: The difference in each method must be tabulated with more technical and numerical results. Also, all papers are discussed with only a paragraph. It must be given some information and visual supplementary figures.

Response: Many thanks. We have introduced additional diagrams and mind maps that graphically describe the articles related to the theme of our work.

The mindmap diagram represented in Figure 1 provides a structured overview of the key concepts and findings from the research paper on BCTs in wearables and IoT. At its core, it highlights the primary topic, "Behavior Change Techniques in Wearables and IoT," branching out into major thematic areas such as the systematic procedures and components of BCTs, the technological aspects of wearables and IoT including sensors, data processing, and personalized feedback, and key study areas including the impact of goal-setting on physical activity, the role of wearable feedback in improving sleep quality, and the effectiveness of gamification in health promotion. Additionally, it touches on various applications of these technologies in health monitoring, chronic disease management, and rehabilitation, while also acknowledging limitations such as challenges in user engagement, data privacy, and addressing diverse populations. The diagram culminates by suggesting future directions in the field, including innovations in technology, personalized health interventions, and considerations for sustainability and compatibility.

In Related work section:

Comment 7: There is no graphical representation of the roadmap of the section.

Response: Thank you very much. With the responses to comments 4 and 6, we have effectively met these well-articulated requirements set forth by the Reviewer.

Comment 8: It is not enough to summarize the literature studies alone. Authors must focus on the architectures or figures of the proposed models of some literature studies.

Response: Thank you very much. We have extensively supplemented the analysis of the literature with new mental schemas. With the responses to comments 4 and 6, we have effectively met these well-articulated requirements set forth by the Reviewer.

Thank you very much.

Sincerely,

Carolina Del-Valle-Soto

Universidad Panamericana. Facultad de Ingeniería. Álvaro del Portillo 49, Zapopan, Jalisco, 45010, México.

Phone: +52 (33) 13682200 | Ext. 4866

Reviewer 2 Report

Comments and Suggestions for Authors

1. What is the research paper's primary objective in exploring behavior change techniques facilitated by wearables and IoT devices?

2. Kindly remove 'e' in line no. 264. (In a 3-arm randomized controlled trial, e authors [ 21] explored the effectiveness of gamification as a behavior change technique for promoting healthy eating habits)

3. Kindly provide the proper captions for the Tables. (Unable to see tables 2.6, 2.7, 2.8..)

4. What is the significance of Figure 7? How is it related to this work?

5. What key behavior change techniques are identified in wearables and IoT devices, as highlighted in the literature review?

6. How does the research analyze trends in behavior change techniques facilitated by wearables and IoT devices over two decades, from 2000 to 2023?

7. What statistical measurements or methodologies are used to understand the evolving landscape?

8. Based on the literature review and case study findings, what are some potential benefits of leveraging wearables and IoT devices for behavior change?

9. Figure 8 & 9 should be enhanced.

10.  For the case study, what about the candidates' age group, any previous medical history, BMI, Health condition? These data need to be presented.

11. How are the parameters (Vital signs) measured with wearable devices?

12. Enhanced statistical analysis is required for the case study data.

13. Kindly avoid 'we' in the manuscript.

14.  Revise the conclusion with research findings obtained from the case study.

15. Add a few recent literatures.

Comments on the Quality of English Language

Moderate editing of the English language is required

Author Response

Dear

Editor

Sensors

We are submitting the paper:

“A Comprehensive Review of Behavior Change Techniques in Wearables and IoT: Implications for Health and Well-being”

Authored by: Carolina Del-Valle-Soto*, Juan Carlos López-Pimentel, Javier Vázquez-Castillo, Juan Arturo Nolazco-Flores, Ramiro Velázquez, José Varela-Aldás and Paolo Visconti.

We would like to thank the reviewers and editors for their detailed analysis of the manuscript; the comments are very valuable to us. In the revised version of the paper, we have incorporated the all changes recommended by the reviewers.

Comments to all observations and suggestions including point-by-point responses are addressed in the following text.

Reviewer 2 comments

Comment 1: What is the research paper's primary objective in exploring behavior change techniques facilitated by wearables and IoT devices?

Response: Many thanks to the Reviewer for his/her invaluable interest in the comments on this manuscript. We have added a clearer paragraph highlighting the specific contribution and objetive of the paper in the Motivation subsection.

This research work makes a significant contribution to the existing literature by addressing a notable gap in the field. Specifically, the scarcity of literature reviews pertaining to wearables and behavior change techniques underscores the novelty and importance of this study. Furthermore, by introducing a new case study to the current body of literature, this review adds valuable insights and empirical evidence to further enrich our understanding of the subject matter. Through meticulous analysis and synthesis of relevant literature, this research not only consolidates existing knowledge but also paves the way for future investigations and advancements in this emerging field. The primary objective of the research paper is to provide a comprehensive review of BCTs as applied within the realm of IoT devices, especially wearables, in the fields of engineering and computer science. This study addresses the increasing integration of technology into daily life and aims to understand the impact of BCTs on user behavior, particularly focusing on their efficacy and implications when implemented in wearable devices. By exploring a case study on the application of BCTs in monitoring vital signs using wearables, the research highlights the profound psychological and social implications associated with technology adoption. It seeks to illuminate how BCTs in wearables can influence user behavior, either positively or negatively. The research makes a significant contribution by filling a notable gap in the literature, offering new insights and empirical evidence, thereby enriching the understanding of the crucial intersection of technology and human behavior.

Comment 2: Kindly remove 'e' in line no. 264. (In a 3-arm randomized controlled trial, e authors [ 21] explored the effectiveness of gamification as a behavior change technique for promoting healthy eating habits)

Response: Many thanks for the typo. Done!

Comment 3: Kindly provide the proper captions for the Tables. (Unable to see tables 2.6, 2.7, 2.8..)

Response: The Reviewer is absolutely correct, and I had indeed overlooked properly captioning those tables. We have corrected all three tables. Thank you very much.

Comment 4: What is the significance of Figure 7? How is it related to this work?

Response: We believe that the impact of citations influences the significance, scientific relevance, and timeliness of the topic we are studying in this review paper. Therefore, we aimed to show the reader that there are Behavior Change Techniques (BCTs) related to wearables that have been receiving increased focus in medical and engineering scientific studies in recent years up to the present date.

Comment 5: What key behavior change techniques are identified in wearables and IoT devices, as highlighted in the literature review?

Response: The Reviewer's question is highly important, and we have emphasized this information at the end of the Related Work section so that readers can more easily understand and summarize the topic. Thank you very much.

As key behavior change techniques, Some of them are identified in wearables and IoT devices, as highlighted in various research articles and literature reviews [5], [64]. The specific BCTs used can vary depending on the device and its intended use, whether it’s for fitness tracking, chronic disease management, rehabilitation, or general health and well-being. Recent research continues to explore how these techniques can be effectively integrated into wearable and IoT technologies to maximize their impact on behavior change. For the most current and specific findings, consulting the latest peer-reviewed articles and systematic reviews in this field is advisable.

  1. Self-Monitoring: This technique involves tracking and recording one’s own behavior, such as physical activity levels, dietary habits, or sleep patterns. Wearables and IoT devices can automate this process, making it more convenient and accurate.
  2. Feedback on Behavior: Devices often provide real-time feedback based on the data collected. This feedback can be about physical activity, heart rate, sleep quality, etc., and is used to encourage positive behavior change.
  3. Feedback on Behavior: Devices often provide real-time feedback based on the data collected. This feedback can be about physical activity, heart rate, sleep quality, etc., and is used to encourage positive behavior change.
  4. Goal Setting: Many wearables and IoT devices allow users to set personal goals related to health and fitness. These goals can be tailored to the individual's current ability and can be adjusted over time.
  5. Social Support: Integration with social networks or community platforms enables users to share their achievements and progress, fostering a sense of community and support.
  6. Rewards and Incentives: To motivate continued use and adherence to health-related behaviors, some devices incorporate reward systems, such as points, badges, or sharing achievements on social media.
  7. Reminders and Alerts: These devices can send reminders or alerts to encourage physical activity, medication adherence, or other health-related behaviors.
  8. Personalization: The ability of these devices to provide personalized information and recommendations based on the behavior and preferences.
  9. Education and Information Provision: Providing users with educational content related to health, wellness, and the benefits of certain behaviors.

Comment 6: How does the research analyze trends in behavior change techniques facilitated by wearables and IoT devices over two decades, from 2000 to 2023?

Response: We are analyzing this information in the Methodology Section, starting from subsection 3.2 Search Outputs and Results. There, we exemplify search methods in the Scopus database and analyze the time period from 2000 to 2023 in searches related to BCTs and wearables. This is in order to provide the reader with an updated context, linking our topic with the scientific impact in engineering and computer science.

Comment 7: What statistical measurements or methodologies are used to understand the evolving landscape?

Response: Thank you very much for the pertinent comment. We have revised and improved the description of our methodology and have also added a step-by-step mind map for greater clarity.

To describe the process of searching for bibliographic references in the context of our research on BCTs in wearables and IoT devices, we follow a systematic and structured methodology. This approach ensures that the literature review is comprehensive, transparent, and replicable. Here is a detailed methodology for the bibliographic search:

Primary Objective: To identify and synthesize existing research on BCTs in wearables and IoT devices. Secondary Objectives: Explore specific aspects like effectiveness, user perception, technological advancements, and application areas. The search was targeted from the year 2000 to 2023 to capture the most recent and relevant studies.

Determine Search Criteria:

  1. Keywords and Phrases: we use specific and relevant terms such as "Behavior Change Techniques," "Wearable Technology," "IoT Devices," "Health Monitoring,".
  2. Inclusion and Exclusion Criteria: we define criteria based on publication year, language, type of article (e.g., peer-reviewed, conference papers, articles), and thematic relevance.
  3. Databases and Sources: we identify suitable databases like Scopus. We consider also specialized journals and conference proceedings in the field.

Conduct the Search: Using the defined keywords, an initial search was conducted in Scopus. This search was broad to ensure capturing a wide range of relevant literature. The initial search results were screened based on titles and abstracts. Papers were selected based on their relevance to the topic, focusing particularly on studies that examined the application and effectiveness of BCTs in wearables and IoT.

Selected articles underwent a full-text review for a detailed understanding of their methodologies, findings, and relevance to our research questions. Statistical Analysis of Search Outputs: The work mentions analyzing the number of documents retrieved, average citations, and distribution of publications across fields like computer science and engineering. This helped in understanding the research landscape and its evolution over the years.

The present review references the use of figures to illustrate the paper selection process, highlight the main keywords in the searches, and show the number of documents per year. These visual tools assisted in better understanding and presenting the trends and patterns in the literature. Throughout the search and review process, gaps in the literature, as well as trends in BCT research within wearables and IoT, were identified. This helped in pinpointing areas that require further investigation.

Key information from each article, such as methodologies, results, and conclusions, was extracted and summarized to support your research objectives. The quality and relevance of each article were assessed to ensure that the most reliable and pertinent information was included in the research.

Comment 8: Based on the literature review and case study findings, what are some potential benefits of leveraging wearables and IoT devices for behavior change?

Response: The question posed by the Reviewer is of utmost importance, and we have added a response paragraph in the Discussion section.

Leveraging wearables and IoT devices for behavior change represents a profound shift in health and behavior management. These technologies, through real-time data capture and analysis, offer a personalized window into individual habits and health metrics, fundamentally transforming the way individuals engage with their health and wellness journeys. By integrating behavior change techniques, these devices not only provide a platform for continuous self-monitoring but also create an interactive, engaging experience that fosters sustained behavioral change. This is more than just a technological advancement; it is a paradigm shift in health management, where individuals are empowered with insights and feedback that make health management proactive, personalized, and responsive. The analytical depth of these technologies lies in their ability to process complex health data, provide contextual feedback, and motivate change through tailored interventions. This convergence of technology and human behavior highlights a future where personalized health management is seamlessly integrated into everyday life, promising significant improvements in health outcomes and overall well-being.

Comment 9: Figure 8 & 9 should be enhanced.

Response: The reviewer is absolutely correct, and we have substantially altered the presentation and analysis of the results to ensure they are truly understandable and contribute meaningfully to the literature on the impact of wearable use in BCTs.

The analysis of the mean, median, and mode seen in Figures 9,10, and 11 reveal intriguing insights into the behavior patterns represented. In the BF dataset, the mean and median values are fairly close, ranging around 15-16, with a slight increase towards the weekend. This suggests a moderate, somewhat uniform level of behavior across the week with a minor peak on weekends. The mode values, however, show more variation, indicating specific frequent behaviors on certain days. The DS dataset has lower mean and median values, averaging around 12-13. This consistency suggests a steady behavior pattern, though the mode values again vary, pointing towards certain predominant behaviors on different days. HR's data exhibit significantly higher mean and median values, around 80, suggesting more intense behavior patterns in this domain. The closeness of these measures indicates a consistent level of behavior, yet the mode suggests variations in the most common behaviors. In OS, the mean and median are remarkably stable across the week, hovering around 94-95, denoting a very consistent behavior pattern. The mode values show minor fluctuations, which might indicate specific recurring behaviors. REMS data show mean and median values in the higher teens, with slight increases on weekends. This implies a somewhat consistent pattern with minor upticks during weekends. The modes, varying around these values, highlight specific prevalent behaviors. T's dataset shows extremely consistent mean and median values, around 37.2-37.3, indicating a very stable behavior pattern across the week. The mode values, closely aligned with the mean and median, reinforce this consistency. Overall, these statistics paint a picture of varying behavioral intensities and consistencies across different domains and days, with some showing uniform behavior throughout the week while others exhibit specific prevalent behaviors on certain days.

Tables 9 to 14 reveal a wide variance in values for each day of the week. Notably, the HR domain exhibits significantly higher variances compared to others, suggesting greater fluctuations in behavioral patterns within this category. The T domain, in contrast, presents relatively low variance, indicating more consistent behavior. Across all domains, there is no clear trend in variance linked to specific days of the week, signifying that the impact of days on behavioral change techniques is not uniform. This diverse range in variances underscores the complexity of human behavior and the challenges in predicting or influencing it through various techniques. Moreover, these findings suggest that the effectiveness of BCTs may vary considerably depending on the day and the specific domain in which they are applied, emphasizing the need for tailored approaches in behavior change interventions.

Comment 10: For the case study, what about the candidates' age group, any previous medical history, BMI, Health condition? These data need to be presented.

Response: The Reviewer is correct, and we had overlooked that important information. We have now added it to the beginning of the case study.

In the case study, the participant profile comprises 30 individuals, carefully selected based on specific criteria to ensure a homogeneous and relevant sample for the research objectives. The age range of these participants is between 20 and 65 years, providing a broad spectrum that encompasses young adults, middle-aged individuals, and those approaching senior age, allowing for a comprehensive analysis across different stages of adulthood. Notably, all participants are non-smokers, a criterion essential for eliminating the potential confounding effects of smoking on energy consumption and vital signs. Additionally, none of the participants have a history of cardiovascular diseases, ensuring that the study's findings are not skewed by underlying cardiac conditions, which can significantly impact both energy consumption and sleep patterns. Furthermore, the participants are not high-performance athletes, aligning the study with a more general population rather than those with exceptional physical conditioning, which could otherwise introduce significant variances in the study's energy consumption and sleep monitoring metrics. It's also important to note that all participants have a normal weight range, as no instances of overweight conditions are reported, ensuring a more uniform baseline for analyzing the impacts of the proposed algorithm on energy consumption and sleep patterns. All participants have provided written informed consent, adhering to ethical research practices and ensuring that they are fully aware of and agreeable to their involvement in the study. This comprehensive and detailed participant profile is crucial for the integrity and applicability of the research findings.

Comment 11: How are the parameters (Vital signs) measured with wearable devices?

Response: Thank you to the Reviewer. We have included this clarification in the case study, as the participants used commercial wearables that they either owned or that we provided to them.

In commercial wearable devices, vital signs such as Breathing Frequency, Deep Sleep, Heart Rate, Oxygen Saturation, REM Sleep, and Temperature are measured through integrated sensors and technology designed for non-invasive and continuous monitoring. Heart Rate is typically measured using optical heart rate monitors that employ light-based photoplethysmography to detect blood volume changes. Sleep stages like Deep Sleep and REM Sleep are discerned via accelerometers and heart rate data, using algorithms to analyze movement and physiological signals during sleep. Oxygen Saturation can be gauged in some advanced wearables using pulse oximetry sensors that estimate blood oxygen levels. Breathing Frequency might be tracked indirectly through movement sensors or algorithms interpreting heart rate variability. For Temperature, wearables often use embedded thermistors or similar components to measure skin temperature, which, while not core body temperature, can provide valuable health insights.

Comment 12: Enhanced statistical analysis is required for the case study data.

Response: The Reviewer is correct, and we have supplemented the case study analysis with the Tables 9 to 14.

Comment 13: Kindly avoid 'we' in the manuscript.

Response: Thank you very much, but we thought that in formal English writing, maintaining the use of the pronoun "we" is often essential for clarity and to establish the appropriate tone, especially in academic or professional contexts.

Comment 14: Revise the conclusion with research findings obtained from the case study.

Response: We have added a substantial portion of conclusions to the final section in order to examine the applications found in the case study as well.

The comprehensive analysis of BCTs across various vital signs, with a focus on mean, median, and mode, underscores the nuanced effectiveness of these techniques in behavior modulation. While each BCT shows efficacy within specific parameters, the variability in results across different days of the week highlights the dynamic nature of human behavior and the complexity of influencing it through systematic interventions. This variability necessitates a tailored approach to behavior change interventions, taking into account individual differences and day-to-day fluctuations. Moreover, the identification of certain BCTs as particularly effective in stabilizing or improving specific vital signs opens avenues for targeted, personalized health strategies. This insight can significantly contribute to the research and development in the science of behavior change, allowing for more precise and effective application of BCTs in health interventions, potentially leading to improved health outcomes and more efficient resource utilization in healthcare and wellness programs. The implications of this study extend beyond the scope of individual health, offering valuable information for the broader field of behavioral science, where understanding and influencing human behavior is central.

Comment 15: Add a few recent literatures.

Response: We have significantly expanded the current literature (as you can see in the newly highlighted references). We have also included a Figure 7 (Mindmap illustrating the interconnections and thematic relationships among key research articles on the impact of wearable technologies in facilitating BCTs) to exemplify the connections of this important literature.

Thank you very much.

Sincerely,

Carolina Del-Valle-Soto

Universidad Panamericana. Facultad de Ingeniería. Álvaro del Portillo 49, Zapopan, Jalisco, 45010, México.

Phone: +52 (33) 13682200 | Ext. 4866

Reviewer 3 Report

Comments and Suggestions for Authors

The article "A Comprehensive Review of Behavior Change Techniques in Wearables and IoT: Implications for Health and Well-being" is devoted to the study of the effectiveness and impact of behavioral techniques created using information technology, in particular wearable devices and Internet of Things (IoT) devices, in the field of engineering and computer science. The review contains 70 literary sources. The authors conducted an extensive review of the relevant literature taken from the Scopus database. The study explores trends over twenty years, from 2000 to 2023. The authors have done a good job by conducting an extensive literature review and identifying trends in the development of behavior change technologies. Below are the comments that will help the authors to finalize the manuscript as follows:

1. Is it an article or a review? There is no experiment in the review, you have it at the end. If this is an article, then it is very large and needs to be shortened. I would recommend dividing the material into 2 articles and here report a review.

2. I think ,,statistical measurements,, are not proper for describing your research. Statistical measurements include determination average value and approximation of dependence of one parameters on other. You visual only result of research number on fig. 6-7.

3. I think that Fig. 6 and 7 should be divided by text. Please, discuss this figs in review including comments of this tends.

4. Fig. 6. It’s hard to conclude a number of articles from this plot. Please give minor grid labels for axis Y.

5. Some tables don’t have a name (line 317, 326, 332)

6. Ask for the permission to use Figure 3.

7. The conclusion could be strengthened by providing clear recommendations for future research directions and practical implications for the field of engineering and computer sciences.

8. Introduction part are very big, please short the first 4 paragraphs.

9. Section ,,1.1 Motivation,, I think it should be added to introduction.

10. Table 4-7. It’s one big table don’t separated them, It would be more convenient for perception to fit it on 2 pages.

11. Figure 8, 9.10 is not clearly visible, you should increase the font size

12. It is recommended to illustrate tables and diagrams, so it will look more interesting and clearer.

This research study stands out from other reviews on the topic due to several key aspects: focus on behavior change techniques in engineering and computer sciences, comprehensive review and analysis, examination of trends over a span of two decades. Overall, these unique aspects distinguish this research study from other reviews on the topic by providing a specialized focus, a comprehensive analysis, a longitudinal perspective on trends to showcase the application of behavior change techniques in technology within the fields of engineering and computer sciences. Thus, it is recommended for publication, but after revision.

Author Response

Dear

Editor

Sensors

We are submitting the paper:

“A Comprehensive Review of Behavior Change Techniques in Wearables and IoT: Implications for Health and Well-being”

Authored by: Carolina Del-Valle-Soto*, Juan Carlos López-Pimentel, Javier Vázquez-Castillo, Juan Arturo Nolazco-Flores, Ramiro Velázquez, José Varela-Aldás and Paolo Visconti.

We would like to thank the reviewers and editors for their detailed analysis of the manuscript; the comments are very valuable to us. In the revised version of the paper, we have incorporated the all changes recommended by the reviewers.

Comments to all observations and suggestions including point-by-point responses are addressed in the following text.

Reviewer 3 comments

Comment 1: The article "A Comprehensive Review of Behavior Change Techniques in Wearables and IoT: Implications for Health and Well-being" is devoted to the study of the effectiveness and impact of behavioral techniques created using information technology, in particular wearable devices and Internet of Things (IoT) devices, in the field of engineering and computer science. The review contains 70 literary sources. The authors conducted an extensive review of the relevant literature taken from the Scopus database. The study explores trends over twenty years, from 2000 to 2023. The authors have done a good job by conducting an extensive literature review and identifying trends in the development of behavior change technologies. Below are the comments that will help the authors to finalize the manuscript as follows:

  1. Is it an article or a review? There is no experiment in the review, you have it at the end. If this is an article, then it is very large and needs to be shortened. I would recommend dividing the material into 2 articles and here report a review.

Response: Many thanks to the Reviewer for his/her invaluable interest in the comments on this manuscript. We understand the concern that the Reviewer has kindly raised. We wish to keep the paper as a Review, however, we have researched and observed that there are reviews which, in the end, include some experimentation that contributes to the state of the art. Therefore, we would like to maintain our case study as our contribution to the search for current literature related to the manuscript's topic.

Comment 2: I think ,,statistical measurements,, are not proper for describing your research. Statistical measurements include determination average value and approximation of dependence of one parameters on other. You visual only result of research number on fig. 6-7.

Response: We have further refined the information filtering to present the relationships in the current literature on wearables, IoT, and BCTs. This improvement is thanks to the relevant feedback from the Reviewer.

Comment 3: I think that Fig. 6 and 7 should be divided by text. Please, discuss this figs in review including comments of this tends.

Response: The Reviewer is correct, and we have added further discussion to the trend analysis of the graphs.

It is evident that certain keywords exhibit dominant patterns over specific years, indicating shifts in research focus and interest. Initially, the concentration was primarily on basic aspects such as 'Social Support' and 'Feedback on Performance,' reflecting early stages of development in this field. As years progressed, notably from 2015 onwards, there was a marked increase in the emphasis on more sophisticated elements like 'Environmental Restructuring' and 'Stress Management or Emotional Control Training.' This shift suggests an evolving landscape where the complexity of studies and applications in wearables and IoT has increased. The year 2023, in particular, shows a significant spike in 'Goal Setting' and 'Social Support,' indicating a resurgence of interest in these foundational aspects, possibly due to new innovations or changes in societal needs. Overall, these trends highlight the dynamic nature of research in this area, adapting to technological advancements and changing priorities. The analysis of citation trends for each of the ten specified keywords reveals distinct patterns in their academic prominence and research interest. 'Social Support' leads significantly with 3,594 citations, indicating its central role and consistent relevance in the field. This is followed by 'Feedback on Performance' and 'Goal Setting,' with 1,035 and 1,025 citations respectively, underscoring their importance in both theoretical and applied research contexts. 'Setting Physical Activity Goals' and 'Self-monitoring of Activity Levels' also show a noteworthy presence with 487 and 35 citations, respectively, reflecting a focused but substantial interest in these areas. In contrast, 'Barrier Identification or Problem-Solving' and 'Environmental Restructuring' have relatively fewer citations, 67 and 19 respectively, suggesting these are emerging or niche areas within the field. Notably, 'Prompt Use of Imagery' has yet to gain traction in the literature, as indicated by zero citations. 'Stress Management or Emotional Control Training' and 'Motivational Interviewing' occupy a middle ground in citation frequency with 60 and 107 citations, pointing to their recognized yet specific application in research. These trends provide valuable insights into the evolving priorities and areas of emphasis within this scholarly domain.

Comment 4: Fig. 6. It’s hard to conclude a number of articles from this plot. Please give minor grid labels for axis Y.

Response: Thank you very much to the Reviewer. We have broken down the data into a table.

Comment 5: Some tables don’t have a name (line 317, 326, 332)

Response: The Reviewer is absolutely correct, and I had indeed overlooked properly captioning those tables. We have corrected all three tables. Thank you very much.

Comment 6: Ask for the permission to use Figure 3.

Response: The figure is entirely of our own creation (made in Matlab), and the data within it have been duly referenced in our manuscript.

Comment 7: The conclusion could be strengthened by providing clear recommendations for future research directions and practical implications for the field of engineering and computer sciences.

Response: We have added a substantial portion of conclusions to the final section in order to examine the applications found in the case study as well.

This comprehensive review underscores the significant advancements and potential of Behavior Change Techniques in wearables and IoT devices, particularly within the realms of engineering and computer sciences. However, to further strengthen the impact and utility of these technologies, it is imperative that future research endeavors focus on developing clear recommendations for their application. Such recommendations should not only include guidelines for effective integration of BCTs in wearables and IoT systems but also elaborate on how these technologies can be tailored to address diverse user needs and contexts. Moreover, future studies should explore the practical implications of these technologies in real-world scenarios, assessing their effectiveness in various settings such as healthcare, sports rehabilitation, and everyday wellness monitoring. This approach will not only contribute to a more profound understanding of the interplay between technology, human behavior, and health but also pave the way for innovative solutions that resonate with the evolving demands of our society. As such, our manuscript provides a foundational platform for further exploration and innovation in the dynamic intersection of engineering, computer science, and health technologies, encouraging researchers and practitioners to delve deeper into this promising field.

The comprehensive analysis of BCTs across various vital signs, with a focus on mean, median, and mode, underscores the nuanced effectiveness of these techniques in behavior modulation. While each BCT shows efficacy within specific parameters, the variability in results across different days of the week highlights the dynamic nature of human behavior and the complexity of influencing it through systematic interventions. This variability necessitates a tailored approach to behavior change interventions, taking into account individual differences and day-to-day fluctuations. Moreover, the identification of certain BCTs as particularly effective in stabilizing or improving specific vital signs opens avenues for targeted, personalized health strategies. This insight can significantly contribute to the research and development in the science of behavior change, allowing for more precise and effective application of BCTs in health interventions, potentially leading to improved health outcomes and more efficient resource utilization in healthcare and wellness programs. The implications of this study extend beyond the scope of individual health, offering valuable information for the broader field of behavioral science, where understanding and influencing human behavior is central.

Comment 8: Introduction part are very big, please short the first 4 paragraphs.

Response: We extend our sincere gratitude to the Reviewer. However, we have endeavored to retain the Introduction as revised by the three reviewers. Since we are presenting the article as a Review paper, we sought to maintain the structure of this section while incorporating additional explanatory diagrams to enhance the understanding of the manuscript's structure.

Comment 9: Section ,,1.1 Motivation,, I think it should be added to introduction.

Response: The Motivation section is, in fact, a subsection of the Introduction. We have deliberately designated it as a subsection to better highlight the contribution of the article and facilitate a more organized reading experience.

Comment 10: Table 4-7. It’s one big table don’t separated them, It would be more convenient for perception to fit it on 2 pages.

Response: The Reviewer is correct, and we have removed the captions that appeared to divide the table. It is indeed a single table spread across four pages, as we have made every effort to condense it; however, any further reduction would render it illegible.

Comment 11: Figure 8, 9.10 is not clearly visible, you should increase the font size.

Response: The reviewer is absolutely correct, and we have substantially altered the presentation and analysis of the results to ensure they are truly understandable and contribute meaningfully to the literature on the impact of wearable use in BCTs.

The analysis of the mean, median, and mode seen in Figures 9,10, and 11 reveal intriguing insights into the behavior patterns represented. In the BF dataset, the mean and median values are fairly close, ranging around 15-16, with a slight increase towards the weekend. This suggests a moderate, somewhat uniform level of behavior across the week with a minor peak on weekends. The mode values, however, show more variation, indicating specific frequent behaviors on certain days. The DS dataset has lower mean and median values, averaging around 12-13. This consistency suggests a steady behavior pattern, though the mode values again vary, pointing towards certain predominant behaviors on different days. HR's data exhibit significantly higher mean and median values, around 80, suggesting more intense behavior patterns in this domain. The closeness of these measures indicates a consistent level of behavior, yet the mode suggests variations in the most common behaviors. In OS, the mean and median are remarkably stable across the week, hovering around 94-95, denoting a very consistent behavior pattern. The mode values show minor fluctuations, which might indicate specific recurring behaviors. REMS data show mean and median values in the higher teens, with slight increases on weekends. This implies a somewhat consistent pattern with minor upticks during weekends. The modes, varying around these values, highlight specific prevalent behaviors. T's dataset shows extremely consistent mean and median values, around 37.2-37.3, indicating a very stable behavior pattern across the week. The mode values, closely aligned with the mean and median, reinforce this consistency. Overall, these statistics paint a picture of varying behavioral intensities and consistencies across different domains and days, with some showing uniform behavior throughout the week while others exhibit specific prevalent behaviors on certain days.

Tables 9 to 14 reveal a wide variance in values for each day of the week. Notably, the HR domain exhibits significantly higher variances compared to others, suggesting greater fluctuations in behavioral patterns within this category. The T domain, in contrast, presents relatively low variance, indicating more consistent behavior. Across all domains, there is no clear trend in variance linked to specific days of the week, signifying that the impact of days on behavioral change techniques is not uniform. This diverse range in variances underscores the complexity of human behavior and the challenges in predicting or influencing it through various techniques. Moreover, these findings suggest that the effectiveness of BCTs may vary considerably depending on the day and the specific domain in which they are applied, emphasizing the need for tailored approaches in behavior change interventions.

Comment 12: It is recommended to illustrate tables and diagrams, so it will look more interesting and clearer.

Response: We have introduced further mental frameworks regarding the key points of our research.

The scheme represented in Figure 1 describes the main points and progression of the research, starting from the revolutionization of daily lives by information technologies, through the introduction of wearables and IoT devices, leveraging behavior change techniques, exploring effectiveness and impact, understanding mechanisms, shedding light on the evolving field, and finally, harnessing the full potential of technologies for positive behavior modifications and improved health and quality of life. Additionally, it includes the classification of wearable applications based on survey results and literature.

Thank you very much.

Sincerely,

Carolina Del-Valle-Soto

Universidad Panamericana. Facultad de Ingeniería. Álvaro del Portillo 49, Zapopan, Jalisco, 45010, México.

Phone: +52 (33) 13682200 | Ext. 4866

Round 2

Reviewer 1 Report

Comments and Suggestions for Authors

All revisions are completed carefully. I think the paper achieved a good impact after my revision comments. The current version can be published by adding the following new IoT reference (I am not one of the authors of this paper): "Farea, A. H., Alhazmi, O. H., & Kucuk, K. (2024). Advanced Optimized Anomaly Detection System for IoT Cyberattacks Using Artificial Intelligence. Computers, Materials & Continua, 78(2)." 

Reviewer 2 Report

Comments and Suggestions for Authors

Congrats to the authors.